# Deep generative computed perfusion-deficit mapping of ischaemic stroke

Chayanin Tangwiriyasakul [1] ✉, Pedro Borges[1], Guilherme Pombo[2], Stefano Moriconi[1],
Michael S. Elmalem[2], Paul Wright [1], Yee-Haur Mah[3], Jane Maryam Rondina[2], Sebastien Ourselin[1],
Parashkev Nachev [2,4] & Manuel Jorge Cardoso[1,4]

Focal deficits in ischaemic stroke arise primarily from impaired perfusion downstream of a critical
vascular occlusion. Though the consequent parenchymal lesion is traditionally used to predict clinical
deficits, the underlying pattern of disrupted perfusion provides information upstream of the lesion,
potentially yielding earlier predictive and localising signals. We previously developed a technique to
compute perfusion maps from routine CT and CT angiography (CTA), an imaging modality widely
deployed in clinical practice and available at large data scales. Analysing computed perfusion maps
(derived from CT and CTA) from 1393 CTA-imaged patients with confirmed acute ischaemic stroke,
here we use deep generative perfusion-deficit inference to localise the neural substrates of NIHSS
sub-scores, explicitly disentangling the distinct topologies of disrupted perfusion and neural
dependence. We show that our approach replicates known lesion-deficit relations *without* knowledge
of the lesion itself and reveals novel neural dependents. The high achieved anatomical fidelity suggests
acute CTA-derived computed perfusion maps may be of substantial clinical and scientific value in rich
phenotyping of acute stroke. By relying only on an imaging modality well-established in the hyperacute
setting, deep generative perfusion-deficit inference could power highly expressive models of
functional anatomical relations in ischaemic stroke within the critical pre-interventional window.

Stroke is worldwide the second most common cause of death and the greatest cause of adult neurological disability[1], rendering even small improvements in its management of great value at the population level. Where its cause is vascular occlusion, the critical pathological process is hypoperfusion with resultant focal ischaemia. The complexity of the cerebrovascular architecture, in interaction with the occlusive mechanism, makes quantification of hypoperfusion challenging. We recently proposed a framework for deriving computed perfusion maps (CPM) from routine clinical CTA data[2]. Our original work showed that CPM showed a high degree of spatial correlation (>0.8) with the time-to-maximum (T-max) map generated from RAPID-AI software analysis of prefusion imaging. Whereas RAPID-AI software requires a dedicated four-dimensional CT (or MR) perfusion scan (4D-CTP), our CPM is derived directly from the routine CTA image, minimising radiation exposure, lowering the barriers to clinical application, and permitting retrospective analysis of historical data.

Perfusion imaging is traditionally used to distinguish infarcted from merely threatened tissue for the purpose of treatment selection[3,4]. But perfusion maps have potentially broader utility in capturing the anatomy of tissue under threat and relating it both to patient outcomes and the relationship between deficits and the underlying neural substrate. Our present study is concerned with illuminating the relation between CPMs and clinical deficits as captured by the NIHSS[5,6], through a novel method of functional anatomical inference—deep perfusion-deficit mapping—based on deep generative spatial inference previously validated in lesion-deficit mapping[7].

The task of lesion-deficit inference requires a model with sufficient expressivity to capture both the topology of the neural substrate and the topology of the lesion-generating pathology. The assumption of no, or simple voxel, interdependence on which conventional mass-univariate approaches are based does not generally hold[8,9], and is manifestly violated in perfusion maps where complex spatial correlations inevitably reflect the complexity of the vascular tree. We need high-dimensional multivariate approaches capable of disentangling the anatomical features of a deficit's neural dependence from those merely induced by the perfusion pattern. Recently, Pombo et al. proposed a deep variational autoencoder-based

¹School of Biomedical Engineering and Imaging Sciences, King's College London, London, UK. ²UCL Queen Square Institute of Neurology, University College London, London, UK. ³King's College Hospital NHS Foundation Trust, Denmark Hill, London, UK. ⁴These authors jointly supervised this work: Parashkev Nachev, Manuel Jorge Cardoso. ✉e-mail: c.tangwiriyasakul@gmail.com; chayanin.tangwiriyasakul@kcl.ac.uk

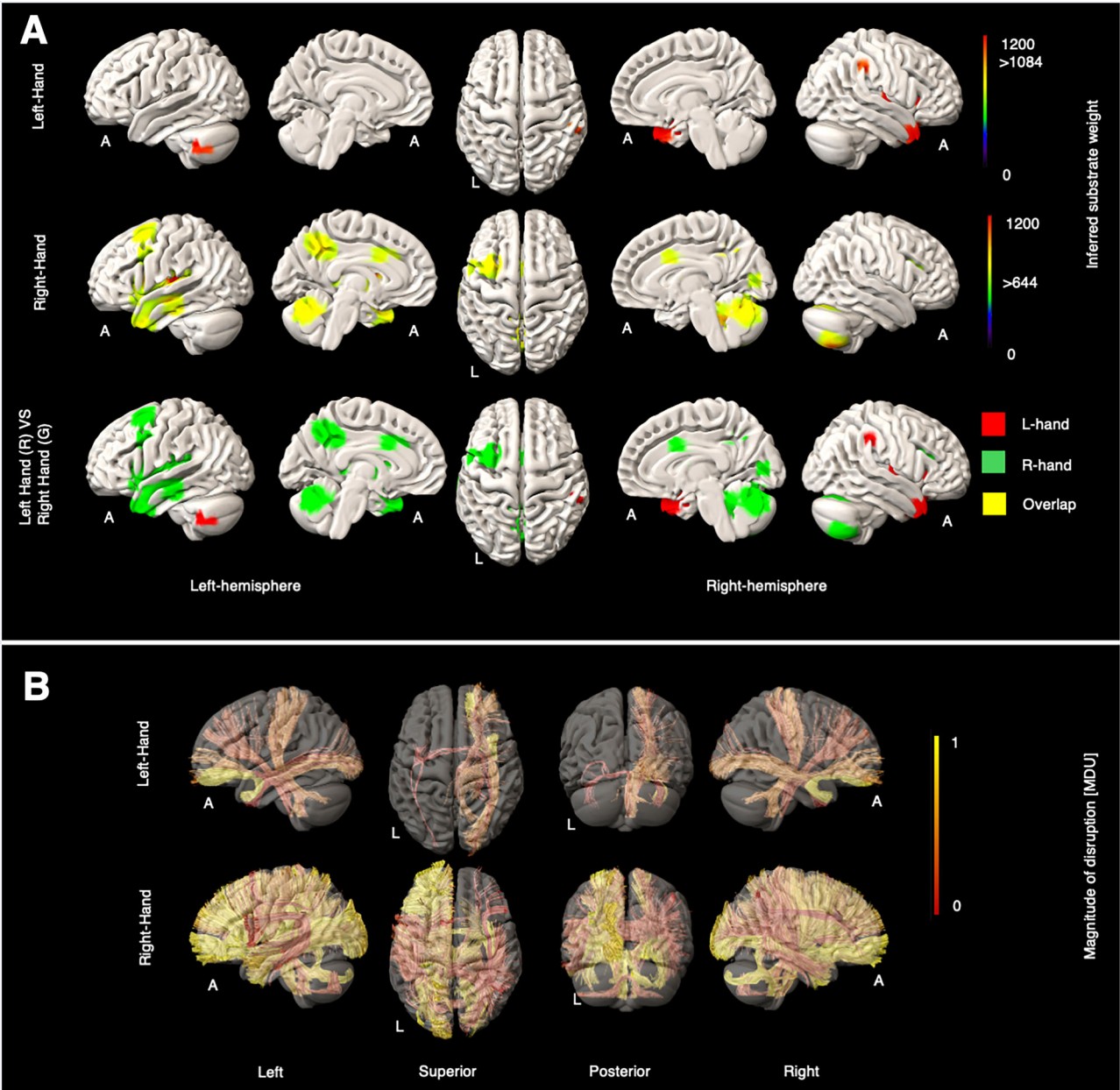

**Fig. 1 | Grey and white matter substrates underlying left- and right-hand motor deficits. A** GM perfusion-deficit maps for the left hand (row 1), right hand (row 2), and their overlap (row 3). In row 1 and row 2, the colour bar indicates the strength of association between each brain area and the NIHSS left-hand and the NIHSS right-hand scores, respectively. Voxels with weights exceeding 1084 (for the left hand) and 644 (for the right hand) are considered significant. In row 3, inferred GM substrates are marked in red for the left hand, green for the right hand, and yellow for their overlap. **B** Disrupted WM tracts for the left hand (row 1) and the right hand (row 2), with the colour bar indicating normalised magnitude of disruption, scaled from 0 to 1 [MDU magnitude disruption unit].

approach—DLM (**D**eep Variational **L**esion-Deficit **M**apping)—for lesion-deficit mapping, based on deriving a joint latent representation of lesions and their associated deficits[7], that in the largest and most comprehensive evaluation of its kind achieves substantially higher fidelity than alternative multivariate and mass-univariate approaches. Deep variational lesion-deficit mapping (DLM) is a generative modelling framework that learns the joint distribution of voxel-wise brain imaging features and behavioural or clinical measures. By modelling the joint distribution of imaging data and scores, it infers spatial patterns most likely to explain observed deficits, while accounting for imaging noise and inter-subject variability. In the present work, we extend DLM to the task of perfusion-deficit mapping in a large cohort of patients with ischaemic stroke ($n = 1393$), inferring the functional anatomy of NIHSS sub-scores from CPMs derived from admission imaging.

## Results

We present our results in subsections clustered by broad behavioural domain: motor, consciousness, gaze and visual, language-and-articulation, somatosensory and attention, and others, see Figs. 1–6. Supplementary Figs. 1–12 provide high-resolution grey (GM) and white matter (WM) maps for each NIHSS sub-score. Supplementary Fig. 13, we summarised the critical tracts for all NIHSS scores, see details in the Supplementary Tables 1–12.

### Motor substrates

Grey matter substrates showed strong lateralisation for both upper and lower limbs (Figs. 1, 2). In the upper limbs, right-sided impairments implicated the contralateral middle frontal gyrus, insula, dorsal cingulate, anterior temporal cortex, occipital cortex, and midline and ipsilateral

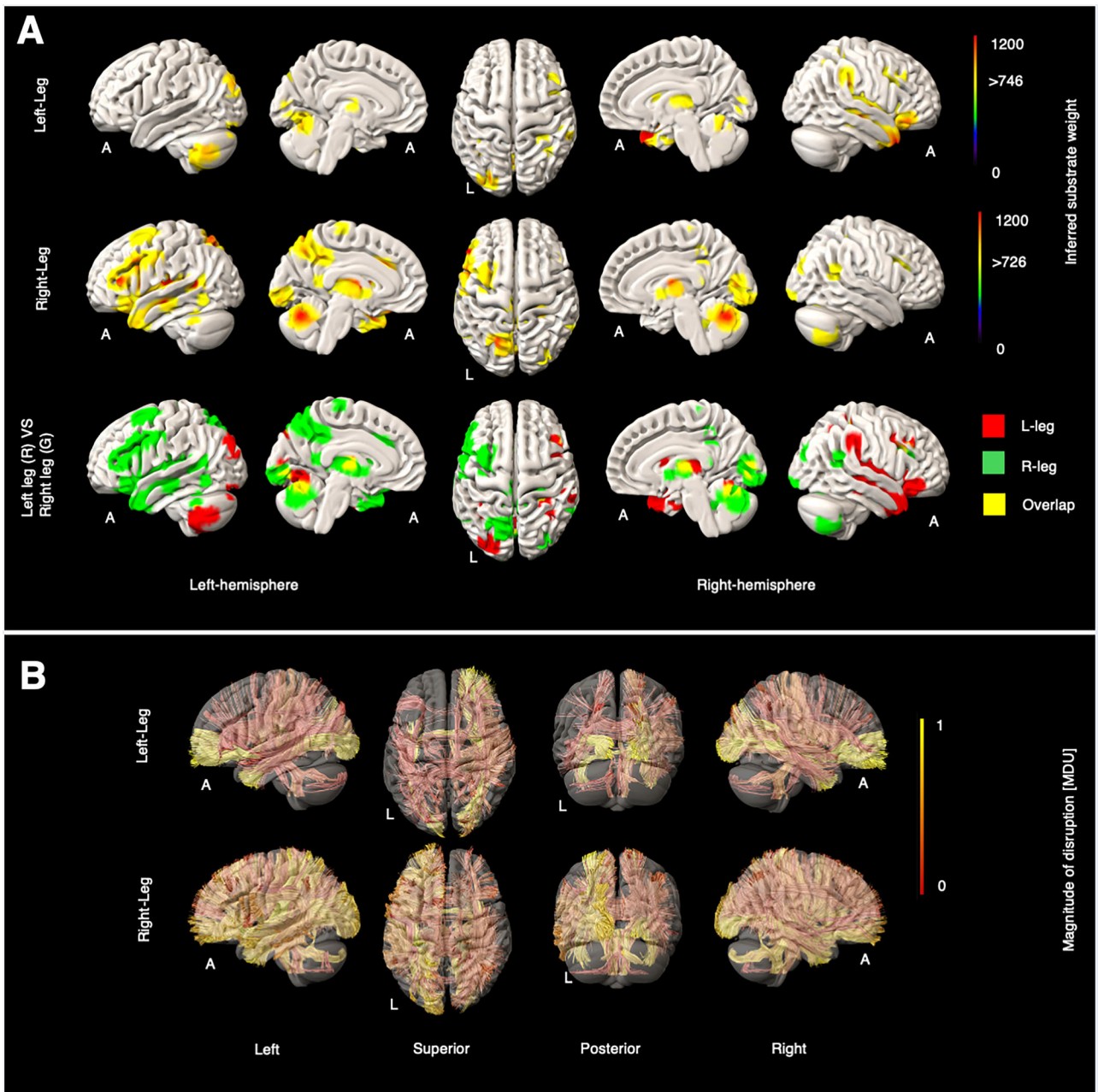

**Fig. 2 | Grey and white matter substrates underlying left- and right-leg motor deficits. A** GM perfusion-deficit maps for the left leg (row 1), right leg (row 2), and their overlap (row 3). In row 1 and row 2, the colour bar indicates the strength of association between each brain area and the NIHSS left leg and the NIHSS right leg scores, respectively. Voxels with weights exceeding 746 (for the left leg) and 726 (for the right leg) are considered significant. In row 3, inferred GM substrates are marked in red for the left leg, green for the right leg, and yellow for their overlap. **B** Disrupted WM tracts for the left leg (row 1) and the right leg (row 2), with the colour bar indicating normalised magnitude of disruption, scaled from 0 to 1 [MDU magnitude disruption unit].

cerebellar regions. Left-sided impairments showed less extensive contralateral cortical associations involving the inferior parietal lobule, insula, and temporal pole, and cerebellar modulation was exclusively ipsilateral. There were no cortical areas of overlap, either supra- or infra-tentorially. In the lower limbs, right-sided impairments overlapped with the upper limb substrates, with the exception of the dorsal cingulate, and additionally implicated the inferior frontal gyrus, parietal cortex, and thalamus, as well as a region of the medial wall close to the foot primary motor cortex. Left-sided impairments were similarly associated with more extensive substrates than in the upper limb, extending to contralateral middle frontal and temporal cortices, as well as thalamus and ipsilateral occipital areas.

The affected WM tracts for the left hand and left leg showed clear contralateral lateralisation, specifically involving the right uncinate fasciculus (UF_R), right dentatorubrothalamic tract (DRTT_R), right inferior fronto-occipital fasciculus (IFOF_R), and right medial lemniscus (ML_R). The WM tracts for the right hand and leg also exhibited lateralisation, though less distinctly. Further details are available in Supplementary Figs. 1–4.

### Consciousness substrates

Loc-question elicited predominantly left-sided GM substrates in the thalamus, insula, inferior and middle frontal gyrus, temporal lobe, inferior

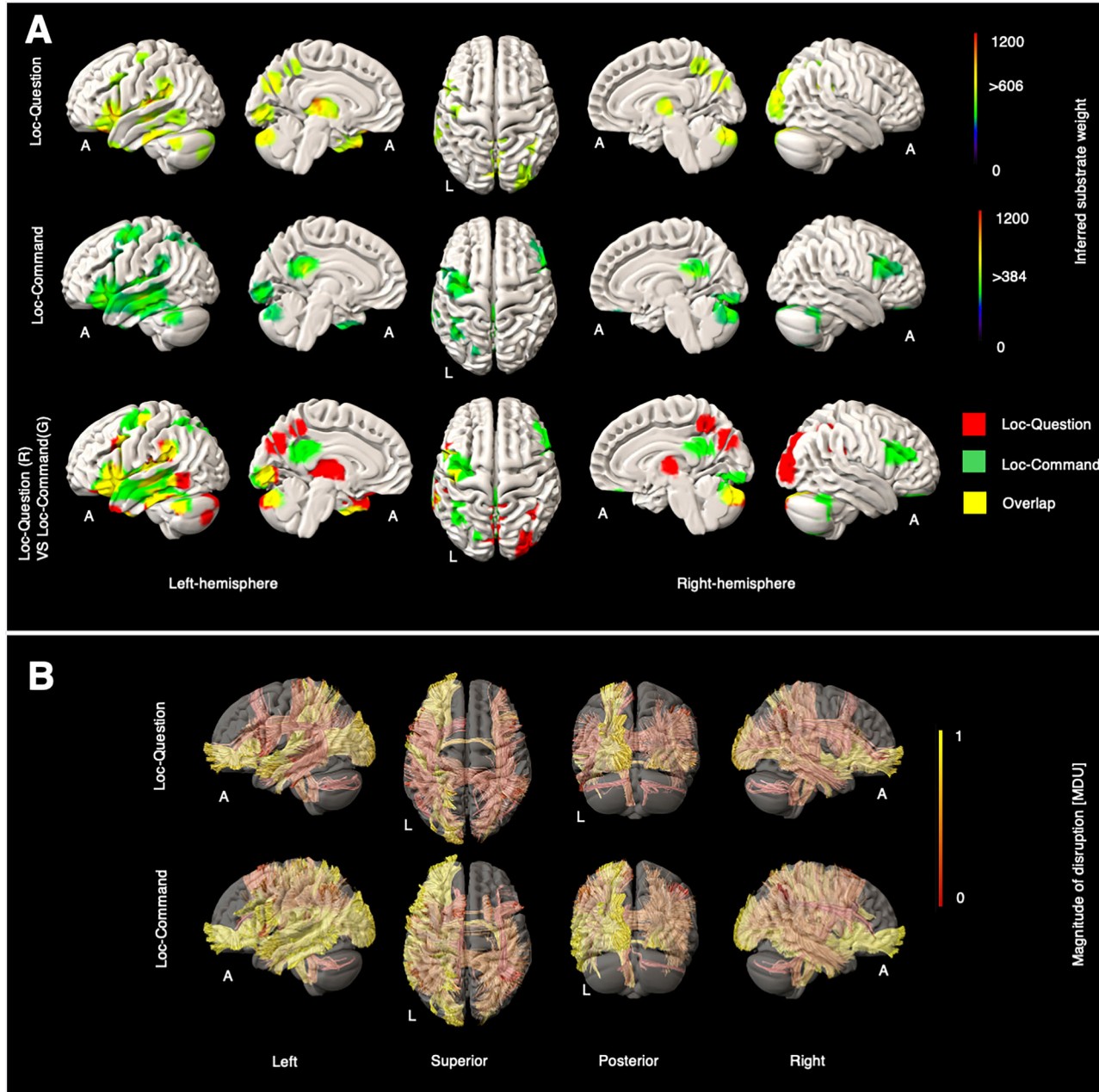

**Fig. 3 | Grey and white matter substrates underlying Loc-question and Loc-command. A** GM perfusion-deficit maps for Loc-Question (row 1), Loc-Command (row 2), and their overlap (row 3). In row 1 and row 2, the colour bar indicates the strength of association between each brain area and the NIHSS Loc-Question and the NIHSS Loc-command scores, respectively. Voxels with weights exceeding 606 (for Loc-Question) and 384 (for Loc-Command) are considered significant. In row 3, inferred GM substrates are marked in red for Loc-question, green for Loc-Command, and yellow for their overlap. **B** Disrupted WM tracts for Loc-Question (row 1) and Loc-Command (row 2), with the colour bar indicating normalised magnitude of disruption, scaled from 0 to 1 [MDU magnitude disruption unit].

parietal lobule, and precuneus, as well as cerebellum and right-sided occipital areas (Fig. 3). Loc-command showed a similar pattern, but now involving left motor cortex, greater modulation of the left temporal lobe, a more ventral part of the precuneus, and right inferior frontal gyrus; thalamic and occipital substrates were not prominent. These patterns cohere with the differential demands on language and motor output of the task. No distinct substrates were identified with Loc-arrival scores.

For both Loc-question and Loc-command, we observed strong left hemisphere lateralisation of affected WM tracts. Nearly all affected bundles in Loc-question were subsets of those in Loc-command, specifically the left middle longitudinal fasciculus (MdLF_L) and left extreme capsule (EMC_L). Motor-related WM tracts, such as the left corticospinal tract

(CST_L) and left dentatorubrothalamic tract (DRTT_L), appeared exclusively or showed greater disruption in Loc-Command than in Loc-question. Further details are available in Supplementary Figs. 5, 6.

## Gaze and visual substrates
Strikingly, gaze elicited bilateral focal frontal and parietal regions closely corresponding to the established localisations of the frontal and parietal eye fields[10,11] (Fig. 4). Visual scores modulated occipital cortex and right inferior frontal regions. In patients with high gaze deficit scores, we observed a greater number of affected WM bundles, particularly in the right hemisphere, with the highest density in the right extreme capsule (EMC_R) and bilateral middle longitudinal fasciculus (MdLF). For visual scores, modest

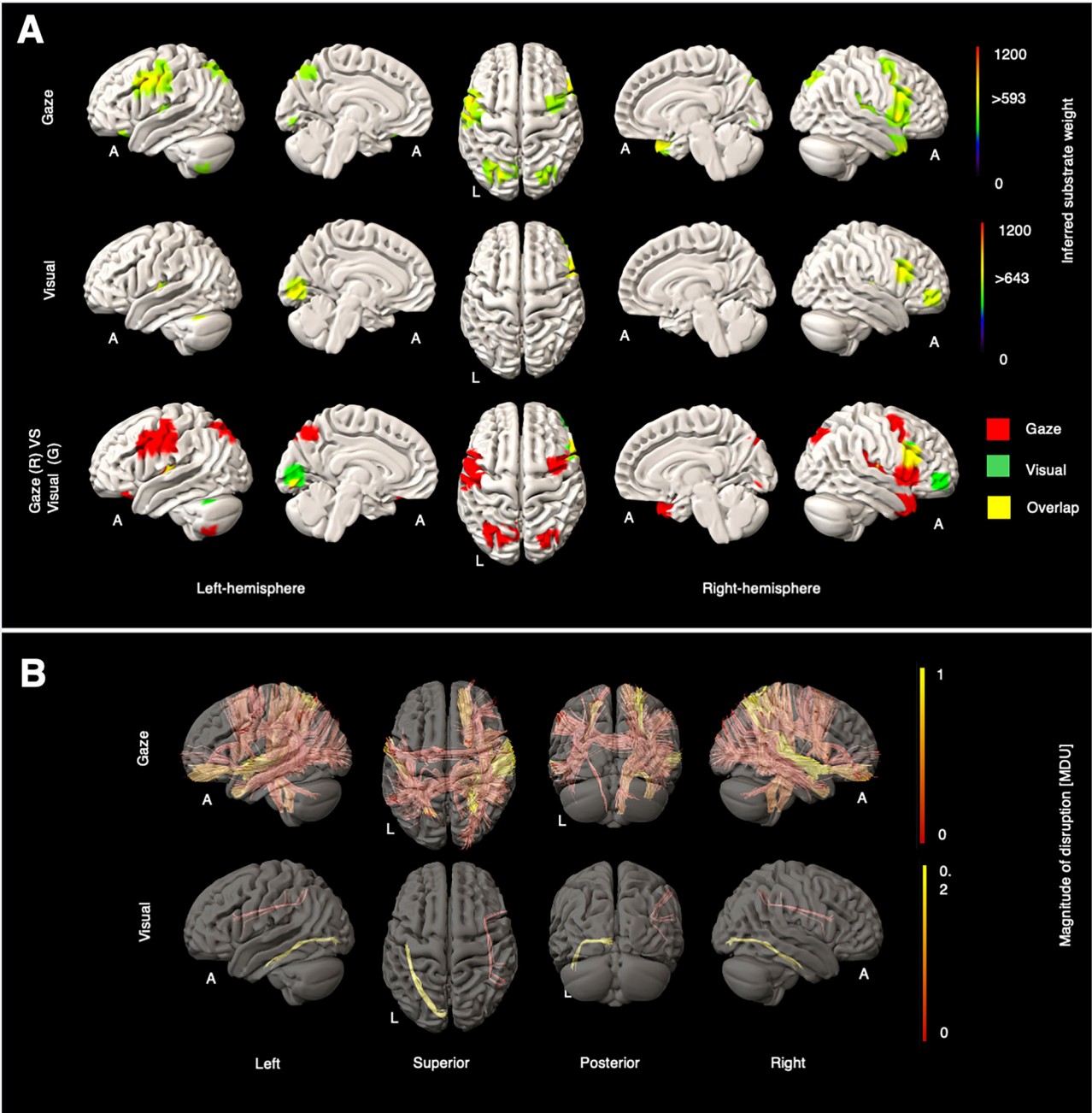

**Fig. 4 | Grey and white matter substrates underlying gaze and visual deficit. A** GM perfusion-deficit maps for gaze (row 1), visual (row 2), and their overlap (row 3). In row 1 and row 2, the colour bar indicates the strength of association between each brain area and the NIHSS gaze and the NIHSS visual scores, respectively. Voxels with weights exceeding 593 (for gaze) and 643 (for visual) are considered significant. In row 3, inferred GM substrates are marked in red for gaze, green for visual, and yellow for their overlap. **B** Disrupted WM tracts for gaze (row 1) and visual (row 2), with the colour bar indicating normalised magnitude of disruption, scaled from 0 to 1 [MDU magnitude disruption unit] for gaze, and scaled from 0 to 0.2 [MDU] for visual.

WM tract disruption appeared in the left inferior longitudinal fasciculus (ILF_L) and right superior longitudinal fasciculus III (SLF3_R). Further details are available in Supplementary Figs. 7, 8.

### Language and dysarthria substrates

Language scores robustly elicited a familiar left-lateralised network involving left inferior frontal, superior and middle temporal, and inferior parietal areas, as well as primary motor cortex, thalamus, precuneus, and cerebellum (Fig. 5). Dysarthria modulated primary motor and inferior and middle frontal areas, and prominently cerebellum, with involvement of precuneus and other posterior regions.

Language deficits were associated predominantly with disrupted left extreme capsule (EMC_L) fibres, followed by the left corticopontine parietal tract (CPT_P_L) and the right extreme capsule (EMC_R). Additionally, both dentatorubrothalamic tracts (DRTT_L/R) showed impairment in both hemispheres. For dysarthria, the right extreme capsule (EMC_R) was most affected, followed by the right middle longitudinal fasciculus (MdLF_R) and then the left extreme capsule (EMC_L). Unlike in language deficits, only mild or no impairment was observed in the left and right dentatorubrothalamic tracts (DRTT_L/R), respectively. Disruption of the vermis tract (V) was elicited only by dysarthria. Further details are available in Supplementary Figs. 9, 10.

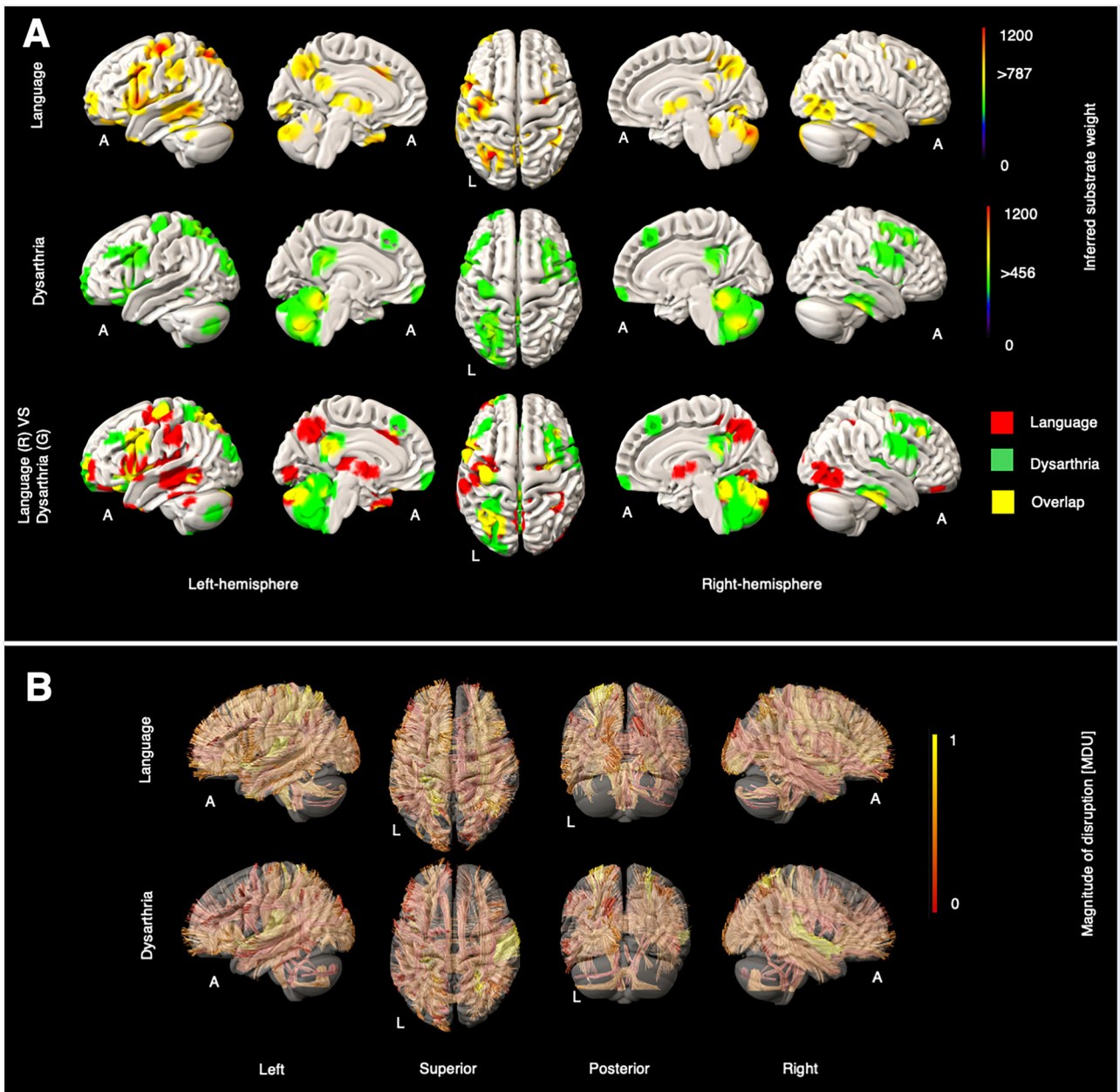

**Fig. 5 | Grey and white matter substrates underlying language deficit and dysarthria. A** GM perfusion-deficit maps for language (row 1), dysarthria (row 2), and their overlap (row 3). In row 1 and row 2, the colour bar indicates the strength of association between each brain area and the NIHSS language and the NIHSS dysarthria scores, respectively. Voxels with weights exceeding 787 (for language) and 456 (for dysarthria) are considered significant. In row 3, inferred GM substrates are marked in red for language, green for dysarthria, and yellow for their overlap. **B** Disrupted WM tracts for language (row 1) and dysarthria (row 2), with the colour bar indicating normalised magnitude of disruption, scaled from 0 to 1 [MDU magnitude disruption unit].

## Somatosensory and attention substrates

Somatosensory scores, inevitably gated by linguistic ability, modulated right inferior frontal, superior temporal, and inferior parietal areas, as well as left occipitoparietal cortex (Fig. 6). Attention, elicited a distributed, predominantly right-sided network, involving inferior frontal, temporal, parietal, and occipital areas, and including areas close to those modulated by gaze.

For somatosensory deficits, disrupted WM tracts were observed in both hemispheres, with a slight predominance in the right. The most affected bundles were the right extreme capsule (EMC_R) and right arcuate fasciculus (AF_R), followed by the right corticopontine parietal tract (CPT_P_R), superior longitudinal fasciculus III (SLF3_R), and medial lemniscus (ML_R). In patients with attentional deficits, disrupted WM

bundles appeared in both hemispheres but were more prominent in the right. The top five affected bundles were the right extreme capsule (EMC_R), right corticopontine frontal tract (CPT_F_R), right middle longitudinal fasciculus (MdLF_R), right corticobulbar tract (CBT_R), and right arcuate fasciculus (AF_R). Further details are available in Supplementary Figs. 11, 12.

## Other NIHSS scores

No substrates were identified for Loc-arrival, facial palsy, and ataxia scores.

## High spatial correlation between CPMs and RAPID T-max maps

We further validated the spatial correlation between CPMs and RAPID T-max in 99 patients. As shown in Fig. 7, CPMs were highly correlated with

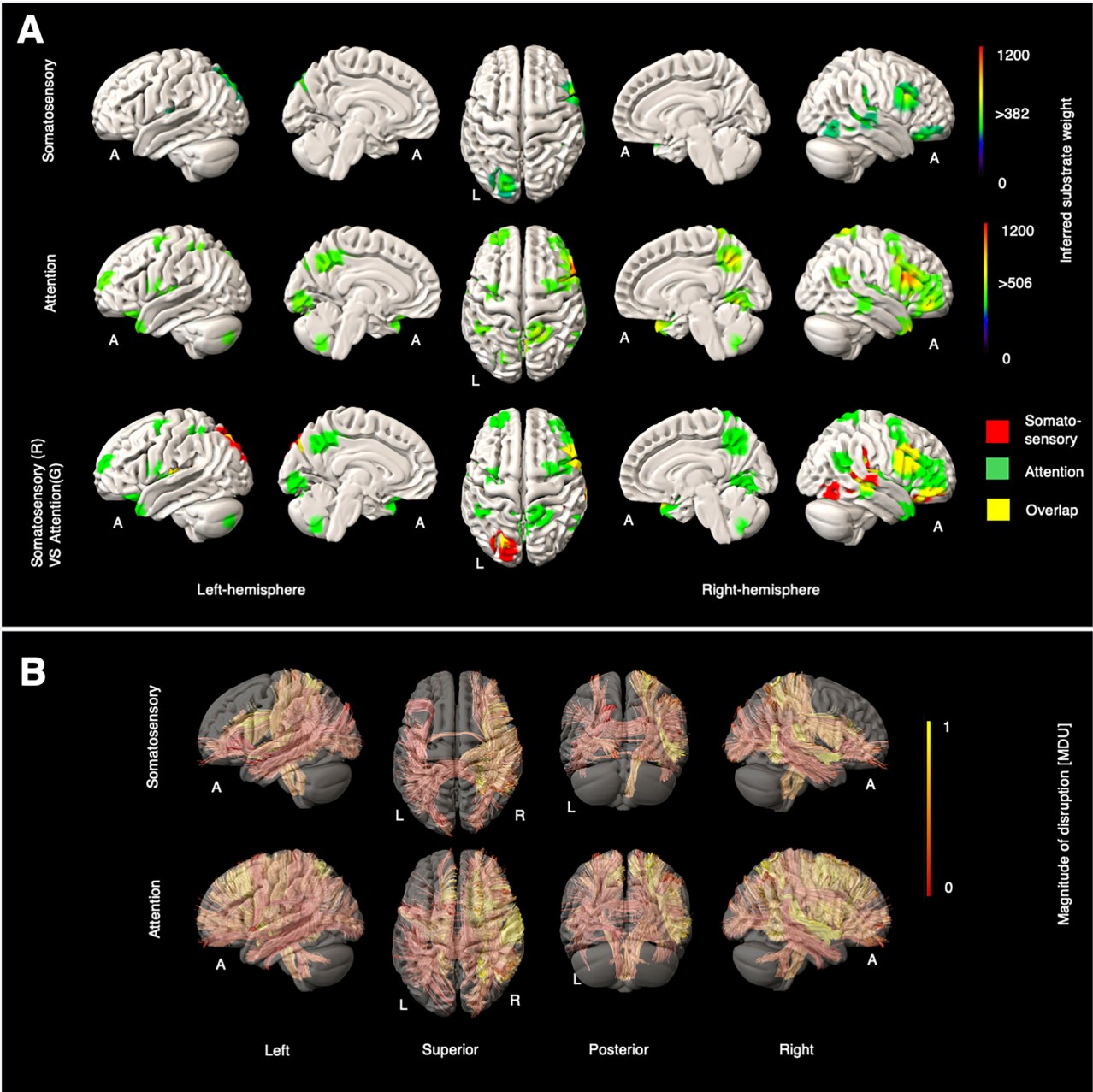

**Fig. 6 | Grey and white matter substrates underlying somatosensory and attention deficits. A** GM perfusion-deficit maps for somatosensory (row 1), attention (row 2), and their overlap (row 3). In row 1 and row 2, the colour bar indicates the strength of association between each brain area and the NIHSS somatosensory and the NIHSS attention scores, respectively. Voxels with weights exceeding 382 (for somatosensory) and 506 (for attention) are considered significant. In row 3, inferred GM substrates are marked in red for somatosensory, green for attention, and yellow for their overlap. **B** Disrupted WM tracts for somatosensory (row 1) and attention (row 2), with the colour bar indicating normalised magnitude of disruption, scaled from 0 to 1 [MDU magnitude disruption unit].

RAPID T-max maps (mean Spearman's $\rho = 0.82$, SD = 0.06; all 99 correlations were statistically significant, $p < 0.05$, see details in the Supplementary Table 13), supporting their validity as an alternative measure of perfusion.

## Discussion

No investigational modality deployable in the acute setting provides a direct index of focal neural dysfunction. Structural imaging reveals tissue changes with largely speculative functional consequences; both BOLD and metabolic signals may be altered by microvascular rather than functional disruption. The neurophysiological impact of focal ischaemia must therefore be inferred from paired observations of anatomical maps of putatively threatened tissue and associated behavioural deficits. The complexities of the brain's functional neuroanatomy and vascular organisation make such inference extraordinarily challenging, for the relation between intricate functional and vascular topologies is illuminated only by suboptimal lesion maps and crudely, reductively indexed behaviour. Crucially, high-resolution acute lesion maps are available exclusively from diffusion-weighted MR imaging that remains challenging to deploy in the clinical—especially hyperacute—setting, severely constraining the range, volume and inclusivity of the data to which models of necessarily high expressivity must be fitted.

Given an anatomical map of threatened tissue in a specific patient, we currently cannot confidently infer the dependence of the associated deficit on the observed topological characteristics, hindering the prediction of both

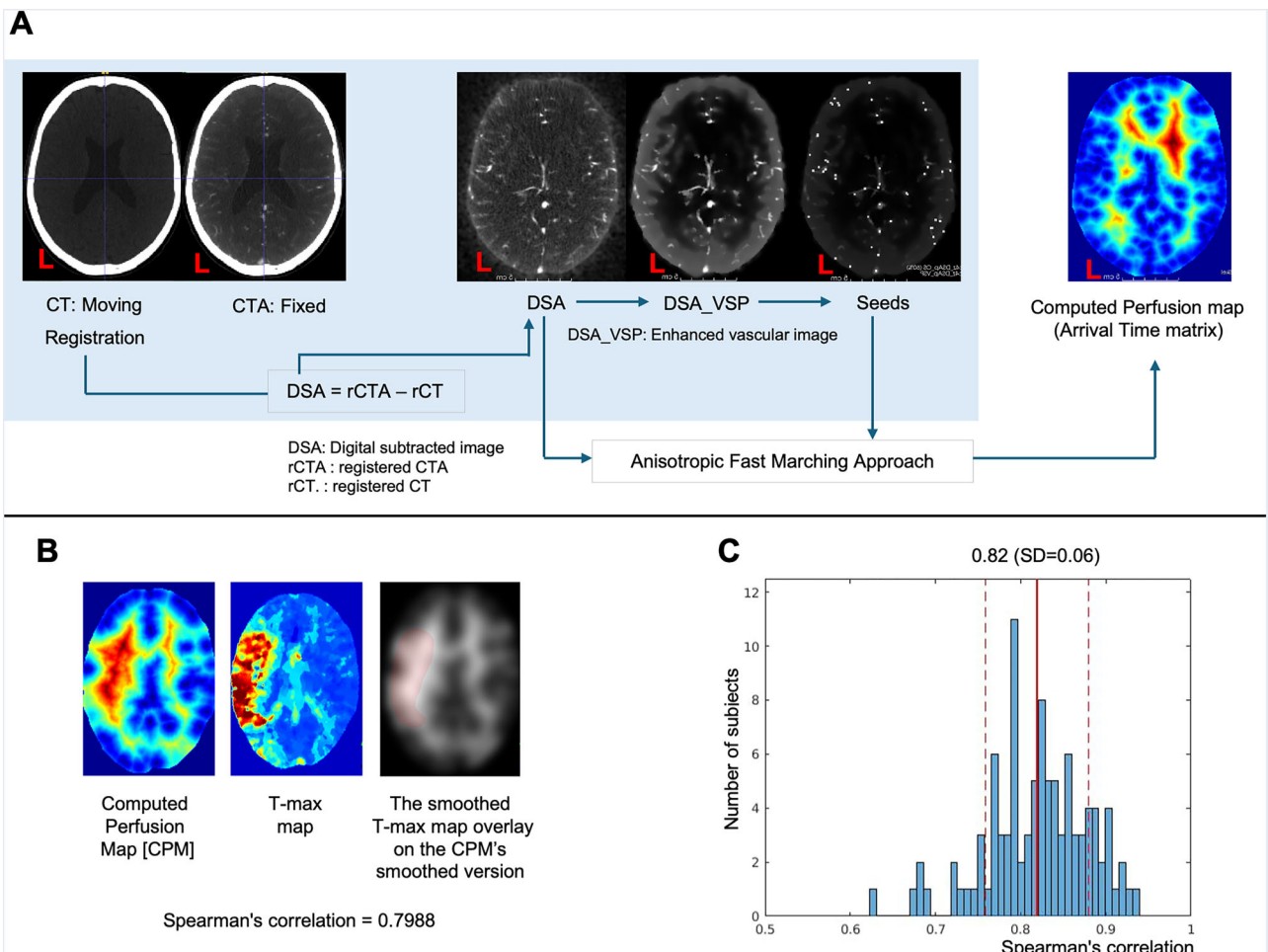

**Fig. 7 | Computed perfusion map generation from CT/CTA and validation against RAPID T-max. A** Preprocessing steps and the estimation of the computed perfusion map (CPM). **B, C** Validation of CPMs against RAPID T-max maps. **B** Example showing spatial correlation ($\rho = 0.7988$) between a computed perfusion map and a RAPID T-max map. **C** Histogram of voxel-wise Spearman correlations across 99 patients (mean $\rho = 0.82$, SD = 0.06, see the Supplementary Table 13 for details). Of these, eight patients were also included in the 1393-patient main analysis; the remaining 91 were outside the main study and lacked complete NIHSS data. This figure supports the use of CPM as an alternative to 4D perfusion imaging in a larger cohort (99 patients). Note that: The left side of the brain is shown to the left.

natural and interventional outcomes. Higher fidelity predictive models require either more informative imaging modalities[12,13] or better analysis of existing signals. Though the former approach is theoretically superior, it is tightly constrained by the difficulty of introducing new modalities into clinical practice and by the time needed to obtain data of sufficient scale. Here, we therefore focus on the latter, exploiting hitherto unrecognised signals in conventional CTA imaging with the aid of computational modelling.

Our approach requires a solution to two critical problems: extracting a tissue-level perfusion map from CTA, and inferring the neural substrate on which the behavioural consequences of impaired perfusion depend. We solve the former by using the anisotropy of the CTA image to derive perfusion maps with minimum prior assumptions on the structure of the underlying signal[14,15]. This approach importantly does not involve any learnt, biological priors that could imperil generalisability beyond a specific training dataset, and involves comparatively few manipulable parameters. We solve the latter by adapting our deep variational lesion-deficit mapping method (DLM)[7]—demonstrated in the largest and most comprehensive evaluation to be substantially superior to all other widely used lesion-deficit methods—to inferring perfusion-deficit relations. DLM directly addresses the central problem of lesion-deficit mapping with vascular lesions: confounding by strong correlations across regions incidentally driven by the structure of the vascular tree. As demonstrated by Pombo et al., this DLM

approach represents the current state-of-the-art among lesion-deficit mapping methods, a property we inherit in its application here.

Applied to a large cohort of patients with acute ischaemic stroke, these innovations enable us to reveal the neural substrates of the wide array of functional deficits captured by NIHSS sub-scores. For functions with well-defined substrates, such as language and gaze control, our results replicate known functional anatomy with surprising spatial precision, supporting the validity of computed perfusion, combined with DLM, as a functionally relevant measure of cerebral vascular integrity. Below, we discuss the neuroanatomical findings and examine the merits, demerits, and prospects of CTA-based perfusion-deficit mapping as a new tool for understanding the critical relation between focal cerebral ischaemia and its functional consequences.

In this study, our main intention is to capture the hyperacute perfusion state (derived mechanistically from CT and CTA). This approach does not account for the temporal evolution of perfusion patterns, such as reperfusion or infarct progression, that occur beyond this window.

The interpretation of inferences to the neural substrates of NIHSS is complicated by the nature of the score and the context in which it is modelled. First, the NIHSS is not designed to isolate cognitive and behavioural functions that are both specific and orthogonal to each other, nor to account for dependencies between them. No simple score-based system could easily handle the censoring of one function by a deficit in another on

which it depends, for example, rule-following by dysphasia. It is therefore inevitable that the inferred neural substrates of a given sub-score will extend to all the functions invoked by the task, which may be common to several sub-scores. This is one important source of the well-known correlations between NIHSS sub-scores that substantially reduce the intrinsic dimensionality of the test[16,17]. Second, as with lesion-deficit mapping in general, the inferred signals are necessarily contrastive against other lesioned patients rather than the normal state. Third, the spatial resolvability of a given substrate will jointly depend on the clinical eloquence of its disruption (i.e., the expressivity of the clinical deficit), the frequency with which it is lesioned, and the size of the lesions that involve it: eloquent substrates commonly affected by small lesions will be more sharply resolved than others.

For motor substrates, clear lateralisation—contralateral above the tentorium and ipsilateral below it—was evident across the four motor-related lesion maps, with more prominent modulation by right-sided deficits. Lateral cortical motor areas were revealed for right-sided deficits, including foot primary motor cortex medially; left-sided deficits induced a similar pattern that did not, however, cross the critical inferential threshold. Inferred WM pathways included corticopontine and dentatorubrothalamic tracts (DRTT) involved in movement generation[18] and speculated to coordinate between the contralateral motor cortex and both ipsilateral[19] and contralateral cerebellum[20]. In contrast to Ferris et al., who found a correlation between motor impairment and the corticospinal (CST) damage[21], we observed only moderately affected corticospinal tract involvement in the right leg WM map. Differences in laterality are potentially related to differences in clinical eloquence, including the association with language disturbance of left hemisphere lesions[22].

For consciousness substrates, Loc-Question and Loc-Command sub-scores invoked left-sided substrates involved in the language processing[23–25] necessary to perform these tasks. Thalamus and precuneus involvement was revealed in the former, in keeping with established evidence for the role of these areas in maintaining awareness[26–29]. Greater thalamic activation in the Loc-question map may underscore the difference between answering questions and following commands, with the thalamus more engaged in the former. Conversely, the motor cortical areas invoked by Loc-Command reflect the demand for movement execution (such as "make a fist, open your hand"); similarly, PCC involvement is consistent with its role in visuomotor integration and executing complex tasks[30,31].

Across white matter substrates, both Loc-question and Loc-command showed clear left lateralisation, likely due to the high language-processing demands of both tasks. The most affected tracts in the Loc-question included the left middle longitudinal fasciculus (MdLF_L)[32,33], left extreme capsule (EMC_L)[34,35], left acoustic radiation tract (AR_L)[36], and left inferior fronto-occipital fasciculus (IFOF_L)[37], each directly or indirectly facilitating language functions. In the Loc-command, these language-related tracts were also observed, along with additional WM tracts related to movement and motor control, such as the left corticopontine tract and left dentatorubrothalamic tract. Additionally, the left arcuate fasciculus (AF_L)[38], which connects Broca's and Wernicke's areas, and the left acoustic radiation (AR_L)[39], which transmits auditory information, were involved in both tasks, highlighting the essential role of language in each.

No neural dependents of Loc-Arrival were evident, potentially owing to inadequate sampling: 73.51% of our patients showed no deficit on this score, the second-highest rate after ataxia.

For gaze and visual substrates, visual sub-scores elicited left-sided occipital cortical substrates, but also right inferior frontal regions. Gaze scores, however, modulated remarkably circumscribed frontal and parietal areas closely corresponding to the known locations of the frontal and parietal eye fields[10,40]. The absence of supplementary eye field substrates is plausibly explained by the comparative rarity of dorsomedial lesions and their modest impact on gaze[41,42]. Other cortical areas, notably inferior frontal regions, precuneus, and cerebellum, fall within well-established networks involved in the control of gaze[43].

Modest modulation of white matter substrates by visual scores—the left inferior longitudinal fasciculus (ILF_L) and the right superior longitudinal fasciculus III (SLF3_R)—is consistent with sparse sampling of visual deficits. The ILF_L connects the occipital and temporal lobes, facilitating information transfer between these regions[44]. SLF3_R is another critical pathway for visuospatial information, linking the right parietal lobe to the right frontal lobe[45]. Gaze modulated the right extreme capsule (EMC_R) and the left and right middle longitudinal fasciculi (MdLF_L and MdLF_R), in keeping with the role of these pathways in visuospatial and attentional processing[46,47].

For language and dysarthria substrates, language sub-scores elicited known frontal and temporal cortical language areas, as well as thalamus, PCC, praecuneus, and midline cerebellum. Dysarthria substrates showed extensive cerebellar involvement alongside primary motor and inferior and middle frontal areas. A degree of overlap is expected, given both tasks impose demands on language, though differing in emphasis. The language test (e.g., describing a picture) is a comparatively complex task requiring high-level semantic comprehension, in which the PCC[48], praecuneus[49] and thalamus[50] play critical roles. Note that the less prominent temporo-parietal activation is predicted by the emphasis on expression (such as describing a picture) over comprehension in the language sub-score. The dysarthria task involves simply reading or repeating sentences, stressing higher-order aspects of linguistic ability to a lesser degree, which plausibly explains the lack of involvement of the thalamus, PCC, and praecuneus. Conversely, extensive cerebellar involvement is shown, consistent with dysarthria as a major manifestation of cerebellar dysfunction[51].

The recruitment of the right frontal lobe seen in the dysarthria map may reflect compensatory mechanisms for cerebellar damage. A study by Geva et al. (2021) using fMRI found that patients with language deficits due to cerebellar damage showed greater activation in the right dorsal premotor cortex (r-PMd) and right supplementary motor area (r-SMA) compared to controls[52]. This compensation may involve recruiting intact brain areas, suggesting an indirect involvement of right motor-related brain areas in speech. Thus, the right motor-related lesions observed in our dysarthria map confirm their vital (though secondary to the cerebellum) roles in speech production.

For both language and dysarthria, we observed critical disruption in the left and right extreme capsule tracts (EMC_L/R). The left extreme capsule (EMC_L) connects Broca's and Wernicke's areas and is considered a vital component of the language pathway[34]. Although EMC_R does not traditionally play a primary role in language generation, a recent DTI study investigating the structural integrity of white matter tracts, including EMC_R, in individuals with primary progressive aphasia suggests a role for right-hemispheric WM, including EMC_R, in supporting language functions[53].

In addition to EMC_L/R, we found critical disruptions in the corticopontine parietal tract, particularly in the left hemisphere (CPT_P_L). Although CPT_P_L is not a core language tract, its role in motor planning and sensory integration may be essential for speech. Three key differences in tract involvement between language and dysarthria were identified: (1) the presence of the vermis tract, (2) the absence of DRTT_L/R, and (3) hemispheric asymmetry between MdLF_L and MdLF_R. These characteristics were uniquely observed in patients with dysarthria. Given its location within the cerebellum and its critical role in motor coordination[54], disruption of the vermis is expected in dysarthria patients.

In the language map, the disrupted dentatorubrothalamic tracts may reflect the high cognitive demands of the NIHSS language tasks, which require thalamic support (seen exclusively in the language GM map). This tract links the motor cortex, thalamus, caudate, and cerebellum[18], underscoring its role in complex motor-cognitive integration.

Both left and right MdLF tracts play significant roles in language production[33]. Similar to the GM map, greater disruption in MdLF_R compared to MdLF_L in dysarthria patients may indicate compensatory activation of right motor-related brain areas to offset cerebellar damage[52], where MdLF_R is involved.

For somatosensory and attention substrates, the somatosensory cortical map primarily included right inferior frontal, superior temporal, and inferior parietal areas, as well as left occipitoparietal cortex. The marked right-sided lateralisation aligns with findings by ref. 55, whose somatosensorially affected cohort was heavily weighted on the right hemisphere.

Attention sub-scores elicited a predominantly right-sided frontoparietal network, consistent with both extant lesion and functional imaging data[56–59]. A degree of overlap with the inferred gaze network is predictable from putatively shared mechanisms for gaze and attentional shifts.

The prominent role of the insula in integrating multimodal sensory information[60] aligns with its appearance in both somatosensory and attention cortical maps. Other shared neural substrates between the somatosensory and attention maps include the right frontal and temporal lobes, which are critical for integrating somatosensory stimuli with cognitive attention networks. Lesions in these areas can lead to somatosensory deficits and neglect by disrupting both somatosensory processing and attentional allocation. Studies by Mort et al. and Dodds et al. have confirmed the importance of the right frontal and temporal lobes in both somatosensory and attention processing[61,62].

In the attention map, we also observed modulation of the praecuneus, the left cerebellum, and the right occipital lobe. The praecuneus has been implicated in spatial cognition[29,63,64]. A study of 80 stroke patients by Verdon et al. found an association between right occipital lesions and neglect[65] in keeping with our observations.

The exclusive left occipital lesion observed in the somatosensory GM map may reflect its secondary contribution to spatial awareness and integration with sensory information, although its primary function is visual. Lesions on the left occipital lobe may not directly cause tactile somatosensory loss but could disrupt visual-spatial integration, potentially affecting overall sensory awareness and spatial interpretation on the contralateral (right) side.

In both the somatosensory and attention maps, affected WM tracts were predominantly located in the right hemisphere. The five most affected tracts in the somatosensory map were the right extreme capsule (EMC_R), right arcuate fasciculus (AF_R), right corticopontine parietal tract (CPT_P_R), right superior longitudinal fasciculus III (SLF3_R), and right medial lemniscus (ML_R). For attention, the five most affected WM tracts were EMC_R, CPT_P_R, the right middle longitudinal fasciculus (MdLF_R), bilateral corticobulbar tracts (CBT_L/R), and AF_R.

The EMC_R, AF_R, and CPT_P_R were common to both maps, in keeping with their joint association with sensory and attention networks[34]. The arcuate fasciculus also plays an important role in orientation and attentional control due to its anatomical connections with the frontal, temporal, and parietal lobes, facilitating sensory integration and attentional processes[38,45]. Lastly, the CPT_P_R connects the parietal cortical regions to the pons and brainstem, supporting sensory and motor integration and contributing to spatial awareness[19,20].

We identified SLF3_R and ML_R as somatosensory-specific tracts. SLF3_R is a white matter tract connecting the supramarginal gyrus and the inferior frontal gyrus, which supports its role in sensory processing, particularly proprioception[66]. A DTI study by Chilvers et al. on 203 stroke survivors found that disconnections in white matter tracts, including SLF III, can lead to somatosensory deficits[67]. ML_R is part of the dorsal column-medial lemniscus pathway, which is a sensory pathway in the central nervous system[68]; thus, lesions in the ML_R tract could disrupt sensory information processing.

The MdLF, a white matter tract connecting the superior temporal gyrus to the posterior parietal cortex, appears in our attention map, in keeping with its role in spatial awareness and attention, particularly in processing verbal-auditory and spatial stimuli[47]. Park et al. suggested that the corticobulbar tract, along with the extreme capsule, facilitates the integration necessary for visuomotor processing[69]. Therefore, disruption of the corticobulbar tract could impact visuomotor orientation and spatial awareness, as observed in NIHSS attention deficits.

No substrates were identified for Loc-arrival, facial palsy, and ataxia sub-scores. The limited power reflects the low entropy of Ataxia and Loc-Arrival (see Supplementary Table 14). For Facial Palsy, an inferred GM substrate over the right frontal lobe was observed, aligning with refs. 70,71, although it was not statistically significant.

The striking correspondence between inferred perfusion-deficit and known functional substrates strongly evidences the fidelity and expressivity of the proposed approach. Given the comparative crudity of NIHSS scores and the challenges of hyperacute imaging, the achieved performance is unlikely to be at its peak. We have deliberately left significant room for algorithmic optimisation of the computed perfusion map, eschewing learnt biological and pathological priors to eliminate trivial memorisation of neuroanatomical patterns during training. However, a correctly specified deep generative model of CPMs, given sufficient data, could enhance initial derivation by constraining the distribution of reconstructed perfusion patterns. Such a model could furthermore incorporate currently unmodelled instrumental parameters, such as the timing-and-phase of contrast administration. Equally, for the same reason the perfusion-deficit model excludes perfusion and functional-anatomical priors and can be extended to a semi-supervised framework utilising external data. That the simple approach here performs so well suggests that high-quality CTA-based perfusion-deficit inference is achievable, illuminating the critical relation between perfusion deficits and their functional consequences in the hyperacute setting. Time efficiency is also critical for potential clinical translation. In our current pipeline, the synthetic perfusion map can be generated from CTA data in an average of 539 s, with subsequent model inference (mapping to NIHSS substrates or reconstructing scores) requiring around 0.25 s per patient. This processing could run in parallel with existing hyperacute stroke workflows, adding minimal delay and without impacting urgent decisions regarding mechanical thrombectomy or thrombolysis.

We now consider seven aspects of the method of relevance to its potential value.

First, the extraordinarily narrow pre-interventional window in stroke —ideally only a few minutes—renders infeasible the other widely available expressive modality, DWI. All extant lesion-deficit mapping studies in the acute setting[70,72–74] use imaging conducted post-thrombolysis or thrombectomy, limiting inferences to the post-interventional state. Thus, differential neural susceptibility to treatment cannot be directly addressed at sufficient data scales or without bias.

Second, though dedicated CT-perfusion imaging optimised for microvascular signals is theoretically superior, it is unavailable at the data scale needed for sufficiently expressive perfusion-deficit models. While multimodal generative models of CTA and CTP could augment either modality, the critical limitation remains the volume of available paired behavioural data. Nonetheless, the tractability of perfusion-deficit inference may promote wider prospective CTP use. Our approach remains the most applicable to large retrospective datasets.

Third, our focus is not merely on quantifying perfusion or relating abnormalities to consequent lesions. Instead, it is on inferring the neural substrate underlying the perfusion-functional deficit relation, which perfusion alone or its correspondence with the final lesion cannot directly indicate. We do not aim to compare computed perfusion abnormalities with CTP (evaluated elsewhere)[2] or with DWI: both are separate questions of independent interest, especially exploring intervention-induced differences. Our focus here is the perfusion-deficit relation, which is disclosed by paired imaging and behaviour. Note that although a full comparison with CTP or DWI is beyond the scope of this work, Fig. 7 presents a targeted validation showing a high spatial correlation (mean Spearman's $\rho = 0.82$) between CPMs and RAPID T-max maps in 99 patients; eight of these were part of the 1393-patient main cohort, while the remainder were outside the main study and lacked complete NIHSS sub-score information.

Fourth, we aim not to predict individual patient deficits but to infer substrates across populations. Like all current lesion-deficit models, our approach assumes a shared underlying functional anatomy for any

behavioural task. Revealing deficit-specific functional substrate distributions could enhance prediction, particularly treatment outcomes, by linking anatomically defined tissue threats with clinical outcomes. Inferred substrates could augment predictive and prescriptive models in observational and RCT settings[75,76].

Fifth, the discriminability of perfusion abnormalities varies with their size in complex ways. Small abnormalities may be concealed by noise; large abnormalities limit the spatial resolution of the inferred substrate. Additionally, vascular transit times vary across the brain[77], with effects on differential susceptibility to vascular insult that remain ill-defined. Any perfusion map, both computed and direct, is liable to resultant variations in fidelity across the brain and patterns of vascular occlusion.

Sixth, though CT is a quantitative modality, instrumental variability may limit generalisability across institutions. Not relying on highly expressive, learnt priors should mitigate the impact of instrumental variation, but future multicentre studies should explore this further.

Finally, perfusion-deficit inference inherits the considerable methodological challenges of lesion-deficit inference, for which only complex models with sufficient expressivity to capture the intricacies of functional and vascular anatomy could conceivably suffice[9,78]. Disentangling functionally critical vs incidental pathological patterns is especially important where, as here, the pathology is spatially highly structured. Since this task lacks a ground truth, the fidelity of any method relies on large-scale semi-synthetic simulations of the kind that support our choice of architecture, shown in the most comprehensive published evaluation to exceed the preceding state-of-the-art by a substantial margin[7]. Nonetheless, this remains an area of intense investigation.

In this study, we analysed NIHSS sub-scores individually rather than modelling latent components, as dimensionality reduction necessarily combines functional domains and can obscure anatomical interpretation. We performed a PCA on the NIHSS sub-scores; the first three principal components (PCs) explained approximately 70% of the variance (Supplementary Fig. 14). As expected, most PCs represented weighted mixtures of multiple NIHSS sub-scores, with lateralisation, overall severity, and language deficits dominating the first three. We visualised the anatomical loadings of this component by multiplying each of the 15 sub-score maps by their corresponding weights and displaying the resultant voxel-wise sum (Supplementary Fig. 14), revealing the expected anatomical lateralisation.

Regarding clinical feasibility and translation, our method is designed to be compatible with routine hyperacute workflows. Computed perfusion maps can be generated from standard CT/CTA in ~10 min using our current (non-optimised) pipeline, with scope for substantially faster processing through parallelisation and code refinement. This enables rapid assessment of cerebral perfusion without additional scanning or radiation exposure. While further prospective validation and appropriate regulatory approval are needed before any clinical deployment is considered, the ability to obtain physiologically interpretable perfusion information from initial imaging may support clinical decision-making in the critical hyperacute management period. CPMs may also allow higher fidelity, quantitative characterisation of the vascular state of the brain in advance of treatment that enables greater precision in estimating treatment effects in the context of interventional trials. Establishing the full range of clinical utility of the approach will require careful exploration of the relations between computed perfusion maps and clinical outcomes, and their modulation by the clinical context, facilitated by the large-scale retrospective data that the widespread use of CTA in stroke management provides.

In conclusion, we introduce a novel perfusion-deficit mapping method combining CTA-derived computed perfusion maps with deep generative topological inference. We validate our approach on a large cohort of patients with acute ischaemic stroke imaged with CTA and assessed on the NIHSS. Inferred substrates across grey and white matter correspond closely to known functional anatomy, while revealing potentially new functional associations. High-quality perfusion-deficit maps from CTA can illuminate neural dependents of clinical deficits in the critical pre-interventional period. Our approach enables quantification of neuroanatomical susceptibility

to ischaemic threat pre-intervention, with wide applications in predictive and prescriptive modelling in cerebrovascular disorders.

## Methods
### Dataset
We retrospectively enroled a cohort of patients admitted to the stroke unit at King's College Hospital between 2015 and 2021, consisting of 2110 patients with a paired CT and CTA on admission. From that cohort, we excluded those with (1) incomplete NIHSS sub-score ($n = 64$), (2) failure on visual inspection of adequate co-registration between CT and CTA ($n = 134$), and (3) presence of a mild deficit as indexed by total NIHSS $\leq 4$ ($n = 519$)[79,80], see details in the Supplementary Fig. 15. The clinical pathway involves obtaining an NIHSS immediately on initial assessment, followed by CT and CTA typically within 30 minutes. Inclusion criteria were met by 1393 patients (mean [SD] age = 69.89 [15.70] years, female = 645, male = 748). Data from King's College Hospital were accessed under ethics protocol 20/ES/0005, approved by the East of Scotland Research Ethics Service for the consentless analysis of irrevocably anonymised data collected in the course of routine clinical care. All ethical regulations relevant to human research participants were followed.

### Data preprocessing and estimation of the computed perfusion map
Figure 7 summarises the data preprocessing step and the estimation of the computed perfusion map. Each patient's CT and CTA were co-registered, and non-linearly normalised with CTSeg (https://github.com/WCHN/CTseg) into MNI space by obtaining the parameters from the CT and applying the estimated deformation field to both images, resliced at $1 \times 1 \times 1$ mm resolution[81]. The normalised images were then processed using the first two steps of VTrails[14,15]: (1) digital subtraction image preprocessing and (2) vascular contrast enhancement and seeds detection. In step 1, we created a digitally subtracted image (DSA) by subtraction of CT from CTA, followed by rescaling of the intensity distribution to the interval 0 to 1. In step 2, we extracted seed points from the vascular contrast-enhanced version of the DSA[14,15]. In this step, we first applied a gradient anisotropic filter to the DSA image as in ref. 82. This filter suppresses noise while preserving the edges of the vascular structure to produce a filtered DSA image or a vesselness speed potential (VSP). We then binarised the VSP image to segment the vessels using the VTrails default threshold of 0.2[14] (https://vtrails.github.io/VTrailsToolkit/). The binarised VSP was then converted into a skeleton image using itkBinaryThinningImagheFilter3D[83] from the Insight Toolkit (ITK; www.itk.org). This step uses a 3D decision tree-based algorithm to thin the binary image to a skeleton representing the centreline of the vascular structure. Every voxel within the skeleton having a value in the VSP image greater than the 75th percentile (VTrails' default parameter) was defined as a seed point, yielding a Seed image. Finally, any seed point located inside the cerebral venous map was removed.

A fast-marching algorithm was applied to the DSA and Seed images to estimate the vascular time-of-arrival at each voxel, with the seed points as the source and the DSA as the speed potential matrix[84]. The fast-marching algorithm is a special case of Dijkstra's algorithm[85]. It computes a minimal geodesic path between two points within an anisotropic medium by minimising an energy function weighted by the medium, in our case, the speed potential matrix[14,85]. We chose seed points located along the centreline of the arterial lumen because the velocity profile is highest along the centreline and monotonically decreases away from it in the orthogonal plane. Our seed points thus represent the source of oxygenated blood to neighbouring brain areas. Using the intensity of the DSA as the speed potential matrix assumes that oxygenation will be faster where the vasculature is denser. Under these constraints, the algorithm calculates, for each voxel outside the skeleton, the fastest propagation time from a skeletal seed point. The resulting voxel-wise image of time-of-arrival comprises the CPM. Since the DSA was normalised between 0 to 1, the CPM is unitless. A higher voxel value reflects a longer time taken for blood to perfuse the location, hence a higher risk of ischaemia at that voxel. Figure 7 depicts all steps to estimate the CPM from CT and

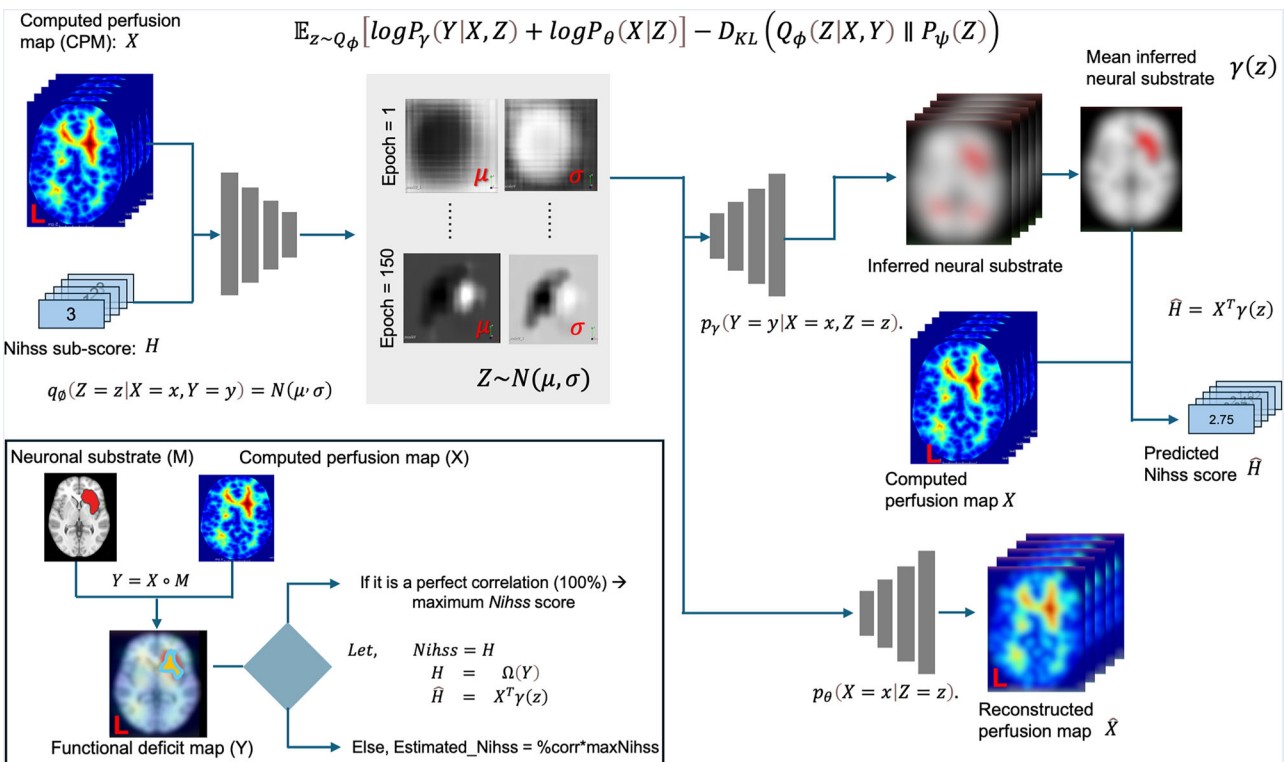

**Fig. 8 | Deep generative computed perfusion–deficit mapping networks.** A five-channel input comprising the computed perfusion map (CPM), X, Y and Z coordinates, and the NIHSS sub-score is encoded into a latent space and decoded through two branches to reconstruct the perfusion map and model the associated clinical deficit.

CTA. We ran our scripts using MATLAB 2020a on an Intel(R) Core(TM) i9-9900K computer. For each patient, it took 539 s (SD = 49 s) to process CT/CTA images, run VTrails and estimate the CPM. After that, all CPMs were resliced into a $2 \times 2 \times 2$ mm image.

**Validating CPMs against RAPID T-max**
To validate our CPMs, we included an additional 91 patients with 4D-CTP acquisitions, in addition to the main cohort. The RAPID-AI software (https://www.rapidai.com/, California, USA) was used to derive a time-to-maximum (T-max) map from each 4D-CTP dataset. T-max is a perfusion parameter representing the time for contrast to reach each voxel from the proximal large artery. Furthermore, eight additional patients from the main cohort also had 4D-CTP scans, yielding a total of 99 patients with available T-max maps. Following the same analysis as in our previous study[2], each exported T-max volume was first converted into a greyscale image and subsequently co-registered to the CPM using SPM12. A Gaussian filter with a kernel size of 10 voxels was then applied to both the exported T-max maps and the CPMs to enhance the visibility of the infarct core[86]. Finally, we calculated Spearman's rank correlation coefficients between the T-max maps and their corresponding CPMs to assess the level of similarity.

**Perfusion-deficit mapping**
We employed deep variational lesion-deficit mapping (DLM), a mechanism for spatial inference, introduced by ref. 7, based on the variational auto-encoder (VAE). The model was fitted to 1393 patients: 90% for training, and 10% equally for validation and calibration.

DLM here learns a latent representation (Z) that explains the joint distribution of CPM and NIHSS sub-scores under the biologically plausible assumption—implicit in all lesion-deficit mapping—that the observed deficit is the dot product between the perfusion map and the inferred neural substrate. The DLM encoder is a convolutional neural net (CNN) mapping the CPM and paired NIHSS sub-score to a 50-dimensional latent representation Z. The DLM decoder is a CNN that maps Z to the inferred neural substrate, in the same space as the CPM, along with a reconstruction of the

CPM. The NIHSS sub-score is also reconstructed, as the dot product between the inferred neural substrate and the original CPM. This is the inductive bias, introduced in ref. 7, that lesion-deficit mapping in general foundationally presupposes. The inset in Fig. 8 depicts the relationship between the underlying neural substrate and the CPM.

**Implementation of perfusion-DLM**
Figure 8 depicts the Perfusion-DLM models implemented in our study. A five-channel matrix comprised of the computed perfusion map (CPM), X, Y, and Z coordinates and the NIHSS sub-score was fed into the network.

The encoder $\varphi$ comprises seven layers. The first six involve the following sequence of operations: (1) a 3D convolution, with a $3 \times 3 \times 3$ kernel and a stride of $1 \times 1 \times 1$; (2) batch normalisation; (3) a nonlinear component-wise activation function (the Gaussian error linear unit function or GELU); and (4) another 3D convolution, with a $2 \times 2 \times 2$ kernel and a stride of $2 \times 2 \times 2$, so that the spatial dimensions are halved after each layer. The number of output channels in each layer is 5, 16, 32, 64, 128, 256, respectively. The seventh layer is a fully connected layer consisting of 100 output channels and 2048 input channels. The first 50 are the mean of the posterior distribution of Z, and the second 50 are its standard deviation.

The decoder consists of two branches, $\gamma$ and $\theta$, which act on separate halves of the 50-dimensional latent representation. The first branch, $\gamma$, maps the first 25 dimensions of Z to the latent substrate; the second branch, $\theta$, maps the second 25 dimensions to the reconstruction of the CPM. Both branches start with a fully connected layer that maps 25 dimensions to $256 \times 2 \times 2 \times 2$. The subsequent layers effectively replicate the encoder, but in reverse. The loss function is the variational lower bound on the log likelihood of the joint distribution of the CPM and the NIHSS sub-score, comprising the sum of (1) the KL-divergence; (2) the L2-loss between the reconstructed and original CPM; and (3) the L2-loss between the reconstructed and original NIHSS sub-score, see Fig. 8.

We minimise the variational lower bound using ADAM, with a learning rate of 1e-4 and a weight decay of 1e-5. The networks were trained for a maximum of 4000 epochs, with a batch size of ten, on an NVIDIA-

DGX-A100. The model weights were saved after each epoch, and after 4000 epochs, we kept the model that best estimates the NIHSS sub-score.

## Calibration of inferred substrates

The outputs from Perfusion-DLM networks are (1) inferred neural substrate, (2) reconstructed perfusion map, and (3) reconstructed NIHSS sub-score. As described in ref. 7, we binarised the neural substrate to determine deficit-associated neural areas by performing calibration on 5% of the data (70 patients), independently selected for each NIHSS sub-score. We searched for the best threshold between a range of 90.5 to 99.5 percentile. At each iteration, we calculated the predicted NIHSS sub-score by multiplying the binarised substrate with a perfusion map for each patient and averaged across all 70 patients from the calibration dataset. The critical level was selected at the iteration with the highest accuracy.

## Tissue-specific inferred substrates

Since the physiological effects of damage to white and grey matter are plausibly different, each derived substrate was segmented into grey and white matter components based on the CTseg white-matter and grey-matter template[81].

For each white matter substrate, we created a tractography of constituent fibre bundles using a white matter bundle atlas[87]. A fibre tract was considered involved if at least two white matter voxels were located within 2 mm of the tract. Note that we performed a sensitivity analysis across eight conditions (2, 4, 6, 8, 10, 12, 14, and 16 WM voxels) and found no notable differences among them. Therefore, two WM voxels were chosen as the criterion for a disrupted WM tract in our study (see Supplementary Table 15). For each bundle, we calculated a magnitude of disruption [MDU], defined as the proportion of disrupted tracts relative to the total streamlines that make up a given specific white matter tract. Unless otherwise specified, all derived substrates in this paper were visualised using SurfICE (https://www.nitrc.org/projects/surfice/).

## Statistics and reproducibility

Spearman's rho correlation was used only for validating computed perfusion maps against RAPID T-max by quantifying voxel-wise associations between the two modalities in 99 patients. All analyses were performed on independent patients ($n = 1393$ in the main cohort; $n = 99$ in the T-max validation cohort, eight of whom overlap with the main cohort). Deep-learning inference does not rely on classical statistical testing; reproducibility is ensured through full methodological and implementation details provided in the Methods and Code availability sections.

Note that: The manuscript is original and was not written by ChatGPT or any other large language model, and no figures were generated using AI tools.

## Report summary

Further information on research design is available in the Nature Portfolio Reporting Summary linked to this article.

## Data availability

The data that support the findings of this study are available from King's College Hospital (KCH), but restrictions apply to the availability of these data, which were used under licence for the current study and so are not publicly available. The data were, however, available upon request and with the permission of KCH.

## Code availability

The custom code necessary to reproduce the lesion-deficit mapping results (including the 3D deep generative lesion-deficit model used in this study) has been deposited in a DOI repository and is available in ref. 88.

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

## Acknowledgements
C.T., P.B., S.M., P.W., J.R., P.N., S.O., and M.J.C. are supported by the Wellcome Trust (WT213038/Z/18/Z). M.J.C. and S.O. are also supported by the Wellcome/EPSRC Centre for Medical Engineering (WT203148/Z/16/Z), and the InnovateUK-funded London AI Centre for Value-based Healthcare. Y.M. is supported by an MRC Clinical Academic Research Partnership grant (MR/T005351/1). P.N. is also supported by the UCLH NIHR Biomedical Research Centre.

## Author contributions
Conceptualisation, C.T., P.N., and M.J.C. Methodology and investigation, C.T., P.B., G.P., S.M., M.E., P.W., Y.M., J.R., P.N., and M.J.C. Resources, P.W., Y.M., S.O., P.N., and M.J.C. Writing C.T., P.N., and M.J.C. Visualisation and analysis, C.T., Y.M., P.N., and M.J.C. Supervision P.N. and M.J.C. All authors revised and approved the paper.

## Competing interests
M.J.C., S.O., and P.N. are founding shareholders in Hologen, a deep generative modelling company. The remaining authors declare no competing interests. A fast-marching algorithm is a part of technologies undergoing patent filing (PCT/GB2020/052368, PCT/GB2021/050349) owned by King's College London.
