## [Transparent Peer Review file · Communications Biology]

Deep generative computed perfusion-deficit mapping of ischaemic stroke

Corresponding Author: Dr Chayanin Tangwiriyaakul

Version 0:

Reviewer comments:

Reviewer #1

(Remarks to the Author)
Esteemed colleagues,

Thank you for the chance to review this interesting manuscript presenting a novel approach to infer neural substrates of clinical deficits in acute ischaemic stroke by combining computed perfusion maps derived from routine CT angiography with deep generative lesion-deficit modelling. Using a large cohort of 1393 patients and a variational autoencoder framework, the authors map NIHSS sub-scores to both grey and white matter substrates without relying on direct lesion imaging, enabling high-resolution functional anatomical inference in the hyperacute pre-interventional window. The manuscript is well written and comprehensive, hence, my comments for the authors to address are mainly minor:

Main comments:

- 1) As far as I understand, the study relied on the synthetic perfusion maps without any further additional validation of these. Even though the previously developed model for this task is showing great correspondence, ($r=0.8$), perhaps the authors can include a comparison/validation with a gold standard perfusion derived from OLEA or RAPID (if the data at hand allows to do so)?
- 2) Given that the NIHSS score is capturing mainly motor deficits and the subscores are highly dependent/intercorrelated, did the authors examine the underlying latent structure of the NIHSS, e.g. with some dimensionality reduction techniques? Mainly to see whether the model outputs remain stable. Would that be feasible to do? Also, because several subscores are showing no substrates, which might be due to little variability.
- 3) The model was validated internally, although the authors discuss this and the difficulty with synthetic perfusion derivation from other less qualitative data, is there a possibility to test this externally, even on a very small external dataset to examine the robustness of the model?
- 4) Is it feasible to test the earlier mentioned latent dimensions and see whether these would show some neuroanatomical or clinical correlation?

Minor adjustments:

Abstract:

- Please specify that the model was trained on synthetic perfusion masks in this sentence in the abstract: "Analysing computed perfusion maps from 1393 CTA-imaged patients ...]"

Intro:

- Great introduction, gives a good overview.
- Line 70, please write out DLM abbreviation at first time use
- Also please include some more information 1-2 sentences about DLM, this would help the clarity

Discussion:

- Line 185-189: DWI is a standard modality that is always acquired in the acute clinical setting
- Discuss a bit in more detail the time it takes to obtain a synthetic perfusion map from CTA that then needs to be fitted into the model and in general the realistic integration of the proposed method into a clinical workflow. In an acute or hyperacute stage, any additional time that is added to the stroke protocol must be carefully balanced with the benefits, given that in that

case, time is brain

- Line 207: superiority of the model implied, but there is no comparison with models or any analyses mentioned before, my apologies if I missed it.

Other: Please include a disclosure statement about the utilization of large language models (chatGPT) in writing this manuscript.

Reviewer #2

(Remarks to the Author)

The overall structure of the paper is quite messy. I don't understand why there are so many sections about the NIHSS substrates. Please simplify them.

The NIHSS score is only one-dimensional. Wouldn't this make the input contain a large amount of redundant information or noise during model training?

The model relies on static imaging (a single time point). Since stroke evolves over time and perfusion patterns change, please discuss this limitation.

NIHSS sub-scores are subjective and coarse measures of function. There may be label noise, and not all deficits map cleanly to anatomical regions.

Version 1:

Reviewer comments:

Reviewer #1

(Remarks to the Author)

I thank the authors for their detailed and constructive responses. The additional validation against RAPID T-max maps convincingly strengthens the manuscript. However, I would strongly encourage that this important validation not be relegated only to the supplementary material but also be described and referenced directly in the main text, since it is central to demonstrating the reliability of the computed perfusion maps. Regarding the external validation issue, I agree with the authors' explanation that suitable datasets are lacking, and I am satisfied with the current approach.

That said, I would still encourage the authors to perform the exploratory analysis of latent NIHSS dimensions that I originally suggested. While I appreciate their concern about interpretability, even a simple dimensionality reduction (e.g., PCA or factor analysis) would provide valuable complementary insights into the underlying structure of the NIHSS sub-scores. This analysis could remain exploratory and not the main focus of the paper, but including it would enhance the robustness and transparency of the reported findings.

Overall, I am satisfied with the revisions and responses, but I would like to stress the suggestion about the inclusion of the RAPID validation in the main text and the addition of the exploratory latent analysis before final acceptance.

Reviewer #2

(Remarks to the Author)

The authors have answered my previous question quite well.

Extra questions:

I am still confused about how the method can be deployed in a clinical setting.

The paper could be further strengthened by presenting more quantitative results.

Version 2:

Reviewer comments:

Reviewer #1

(Remarks to the Author)

Thank you for addressing my concerns. I am satisfied with the integration of the raised questions and have nothing more to add.

Reviewer #2

(Remarks to the Author)

The authors generally address my concerns, but ultimately we still need stroke clinicians to be involved. How to optimize collaboration between clinicians and this model remains an important issue that should be addressed in future work.

Overall feedback

We would like to thank both reviewers and the editorial team for their time and effort.

Reviewers' comments:

Reviewer #1 (Remarks to the Author):

Esteemed colleagues,

Thank you for the chance to review this interesting manuscript presenting a novel approach to infer neural substrates of clinical deficits in acute ischaemic stroke by combining computed perfusion maps derived from routine CT angiography with deep generative lesion-deficit modelling. Using a large cohort of 1393 patients and a variational autoencoder framework, the authors map NIHSS sub-scores to both grey and white matter substrates without relying on direct lesion imaging, enabling high-resolution functional anatomical inference in the hyperacute pre-interventional window. The manuscript is well written and comprehensive, hence, my comments for the authors to address are mainly minor:

Main comments:

1) As far as I understand, the study relied on the synthetic perfusion maps without any further additional validation of these. Even though the previously developed model for this task is showing great correspondence, ($r=0.8$), perhaps the authors can include a comparison/validation with a gold standard perfusion derived from OLEA or RAPID (if the data at hand allows to do so)?

Answer

We thank the reviewer for this suggestion. To further validate the synthetic perfusion maps used in our study (termed "computed perfusion maps"), we extended our prior analysis [Tangwiriyasakul et al., 2024] by comparing the computed maps with RAPID-derived T-max maps in an independent dataset of 99 subjects from the cohort in Tangwiriyasakul et al, 2024. Note that: among these, 8 subjects overlap with our 1393-patient cohort reported in the main manuscript.

As shown in Figure 1, we observed a consistently high voxel-wise correlation (mean Spearman's $\rho = 0.82$, $SD = 0.06$), confirming the robustness of the synthetic perfusion estimation. These results replicate and extend our earlier conference findings.

We have not included this figure in the main manuscript for two main reasons:

1. The validation dataset (91/99 subjects) was recorded after 2019, outside the scope of our main study period.
2. NIHSS scores were incomplete for these subjects, and the data were collected during the COVID-19 period, which introduced other clinical and operational confounds.

However, to strengthen our claim previously presented in Tangwiriyasakul et al. (2024) in a larger cohort (99 subjects), we have included the following sentences "Note that although a full comparison with CTP or DWI is beyond the scope of this work, Supplementary Fig. 15 presents a targeted validation showing a high spatial correlation (mean Spearman's $\rho = 0.82$) between CPMs and RAPID T-max maps in 99 subjects; eight of these were part of the 1,393-patient main cohort, while the remainder were outside the main study period and lacked complete NIHSS sub-score information." on page 16 in the main manuscript; and we have added this figure in the supplementary information (Supplementary Figure 15).

Figure 1 (Supplementary Figure 15): Validation of synthetic perfusion maps against RAPID T-max map

A: Typical example shows spatial correlation (0.7988) between a computed perfusion map and a RAPID T-max map B: Histogram of voxel-wise Spearman correlations across 99 subjects (mean $\rho = 0.82$, SD = 0.06). Of these, 8 subjects were also part of the 1393 patients in our main analysis. The remaining 91 subjects were recorded after 2019, outside the main study window and without complete NIHSS data.

Reference

Tangwiriyasakul, C. et al. Framework to Generate Perfusion Map from CT and CTA Images in Patients with Acute Ischemic Stroke: A Longitudinal and Cross-Sectional Study. in 154–162 (2024). doi:10.1007/978-3-031-76160-7_15.

2) Given that the NIHSS score is capturing mainly motor deficits and the subscores are highly dependent/intercorrelated, did the authors examined the underlying latent structure of the NIHSS, e.g. with some dimensionality reduction techniques? Mainly to see whether the model outputs remain stable. Would that be feasible to do? Also, because several subscores are showing no substrates, which might be due to little variability.

Answer:

We agree with the reviewer that the NIHSS exhibits a latent structure, as individual sub-scores are not fully independent; for example, right upper and lower limb motor scores are often correlated, as are language-related items. This reflects both shared vascular territories and functional interdependencies.

Our aim, however, was not to model or predict latent components of the NIHSS, but to examine each sub-score separately to identify neuroanatomically and functionally interpretable substrates. Latent components derived from dimensionality reduction may mix multiple functional domains, making their anatomical interpretation difficult. Since each sub-score was predicted independently, without conditioning on the other sub-scores, the presence of correlation between sub-scores ought not to influence the inferred maps or destabilise the inferential models. The derived maps will, of course, show such topological similarities as shared substrates between the functions tested by the sub-scores entail.

3) The model was validated internally, although the authors discuss this and the difficulty with synthetic perfusion derivation from other less qualitative data, is there a possibility to test this externally, even on a very small external dataset to examine the robustness of the model?

Answer

Unfortunately, we do not have an external validation dataset with complete NIHSS sub-scores suitable for full model testing. However, our study cohort is a large dataset of relatively unselected patients, and we performed internal testing on a dedicated 5% calibration set that was held out entirely from both training and hyperparameter validation. This set was used only after the model was finalised, providing an independent internal test of robustness.

Secondly, the computed perfusion map (CPM) model does not involve any trainable parameters, so the possibility of data-driven overfitting is essentially eliminated. While the DLM model does involve training, the downstream relationship between CPMs and the inferred anatomy is not explicitly learned but instead inferred from the observed behaviour. One of the most striking aspects of the results is the close correspondence between the inferred anatomy and the known functional anatomy of the brain; patterns for which the model has no prior knowledge.

As mentioned in the response to the first question, we have access to a separate external dataset of 91 CTA-imaged patients collected after the main study period. Unfortunately, this dataset lacks NIHSS sub-scores, which limits its immediate use for full model inference and lesion mapping.

4) Is it feasible to test the earlier mentioned latent dimensions and see whether these would show some neuroanatomical or clinical correlation?

Answer

It is possible to test the latent NIHSS dimensions; however, we believe that such results would be hard to interpret. At present, there is no clear evidence from the literature regarding the anatomical substrates underlying these latent components. Our objective in this study is instead to demonstrate the coherence between the functional-anatomical maps derived for interpretable clinical functions and existing functional-anatomical knowledge. Nonetheless, we agree that exploring latent dimensions could be an interesting direction for future work, particularly if supported by emerging evidence on their anatomical correlates.

Minor adjustments:

Abstract:

5) Please specify that the model was trained on synthetic perfusion masks in this sentence in the abstract: “Analysing computed perfusion maps from 1393 CTA-imaged patients ...[.]”

Answer:

We thank the reviewer. We have added “(derived from CT and CTA)” to clarify the data source and emphasise that the perfusion maps were computed rather than directly acquired.

Intro:

6) Great introduction, gives a good overview.

Answer: We thank the reviewer for the positive feedback on the introduction.

7) Line 70, please write out DLM abbreviation at first time use

Answer

We thank the reviewer and apologise for the omission. We have added “**Deep Variational Lesion-Deficit Mapping**” after DLM at its first introduced.

8) Also please include some more information 1-2 sentences about DLM, this would help the clarity

Answer

We thank the reviewer and this essential suggestion to clarify the communication between us and the readers. We have added “Deep Variational Lesion–Deficit Mapping (DLM) is a generative modelling framework that learns the joint distribution of voxel-wise brain imaging features and behavioural or clinical measures. By modelling the joint distribution of imaging data and scores, it infers spatial patterns most likely to explain observed deficits, while accounting for imaging noise and inter-subject variability.” on page 3.

Discussion:

9) Line 185-189: DWI is a standard modality that is always acquired in the acute clinical setting

Answer

We thank the reviewer. We agree that DWI is an important and widely used modality in acute stroke. However, in our study centre (King’s College London), DWI is not routinely acquired during the hyperacute phase for all patients. When it is performed, it typically occurs around 24 hours after admission; well beyond the time window targeted by our study, which focuses on pre-interventional imaging. Consequently, we relied on CTA-derived computed perfusion maps, which are available at presentation and enable analysis during the critical hyperacute period.

10) - Discuss a bit in more detail the time it takes to obtain a synthetic perfusion map from CTA that then needs to be fitted into the model and in general the realistic integration of the proposed method into a clinical workflow. In an acute or hyperacute stage, any additional time that is added to the stroke protocol must be carefully balanced with the benefits, given that in that case, time is brain

Answer

We fully agree with the reviewer that time efficiency is critical in acute stroke care. We have added the following sentences in the discussion section (on page 15 at the Deep perfusion-deficit mapping subsection):

“Time efficiency is also critical for potential clinical translation. In our current pipeline, the synthetic perfusion map can be generated from CTA data in an average of 539 seconds, with subsequent model inference (mapping to NIHSS substrates or reconstructing scores) requiring around 0.25 seconds per patient. This processing could run in parallel with existing hyperacute stroke workflows, adding minimal delay and without impacting urgent decisions regarding mechanical thrombectomy or thrombolysis.”

11) Line 207: superiority of the model implied, but there is no comparison with models or any analyses mentioned before, my apologies if I missed it.

Answer

We thank the reviewer for this observation. The term “superiority” originates from Pombo et al. (2023), who explicitly compared their DLM approach with other widely used lesion–deficit mapping methods and reported substantially higher fidelity. To avoid confusion, we have added “As demonstrated by Pombo et al., this DLM approach represents the current state of the art among lesion–deficit mapping methods, a property we inherit in its application here.”, see page 7.

12) Other: Please include a disclosure statement about the utilization of large language models (chatGPT) in writing this manuscript.

Answer:

We have included the sentence “The manuscript is original and was not written by ChatGPT or any other large language model, and no figures were generated using AI tools.” at the end of the manuscript, see page 17 after conclusion section.

Reviewer #2 (Remarks to the Author):

1) The overall structure of the paper is quite messy. I don't understand why there are so many sections about the NIHSS substrates. Please simplify them.

Answer:

Thank you for your feedback. The goal of this study is to investigate each of the 15 NIHSS sub-scores to create interpretable functional–anatomical maps. As such, we have presented the results in six subsections: (1) motor substrates (hand and leg motor scores), (2) consciousness substrates (including LOC-question and LOC-command scores), (3) gaze and visual scores, (4) language and dysarthria substrates, (5) somatosensory and attention (including sensory and inattention NIHSS scores), and (6) others (including the scores without significant substrates, please see “*Negative sub-scores*” at the discussion section of the manuscript for details).

This grouping is designed to focus on each function (such as motor-related function or language and speaking). Our intention is to provide a comprehensive view of brain-behaviour relationships across the full range of NIHSS items, which is why these were detailed individually rather than using the total NIHSS score (a sum of the 15 sub-scores).

2) The NIHSS score is only one-dimensional. Wouldn't this make the input contain a large amount of redundant information or noise during model training?

Answer

As explained earlier, the intention of this study is to analyse each NIHSS sub-score separately to create functional–anatomical maps, rather than using the total NIHSS score. This approach captures the functional anatomy of distinct behavioural domains. Applying DLM to the computed perfusion maps, we obtained results that show a close correspondence between the inferred anatomy and the known functional anatomy of the brain-patterns for which the model has no prior knowledge. This shows that both the CPMs and the sub-scores are informative.

3) The model relies on static imaging (a single time point). Since stroke evolves over time and perfusion patterns change, please discuss this limitation.

Answer

We thank the reviewer for this point. All CTA data in this study were acquired at a single acute time point in the arterial phase, typically capturing the MCA and its downstream branches, which are involved in 85–90% of strokes. We acknowledge that perfusion patterns evolve over time and that static, single time point imaging might not fully capture this dynamic process. In our cohort, imaging was obtained within the hyperacute phase (10–20 minutes after admission), which does not always reflect later reperfusion, infarct progression, or secondary events. Our aim was to characterise functional–anatomical mapping specifically within this early window, to demonstrate the richness of computed perfusion maps mechanistically derived from routine CT and CTA. Nevertheless, we have added the following statement in the Discussion section (see page 8).

“In this study, our main intention is to capture the hyperacute perfusion state (derived mechanistically from CT and CTA). This approach does not account for the temporal evolution of perfusion patterns, such as reperfusion or infarct progression, that occur beyond this window.”

4) NIHSS sub-scores are subjective and coarse measures of function. There may be label noise, and not all deficits map cleanly to anatomical regions.

Answer

We agree that NIHSS sub-scores, while clinically validated, are subject to measurement variability and do not capture all aspects of function. However, they remain the most widely used clinical scale in acute stroke care and are routinely collected at the bedside, making them well-suited for large-scale modelling. The associated noise is expected to be approximately Gaussian. Our DLM model minimises the loss function of the variational lower bound on the log-likelihood of the joint distribution of the CPM and the NIHSS sub-score. With a large cohort such as ours (1,393 subjects), the effect of subjective bias or label noise is expected to be modest.

We are grateful to the editor and reviewers for their thoughtful and constructive feedback, which has helped us to improve the clarity, transparency, and clinical relevance of the manuscript.

Reviewer #1 (Remarks to the Author):

Q1: I thank the authors for their detailed and constructive responses. The additional validation against RAPID T-max maps convincingly strengthens the manuscript. However, I would strongly encourage that this important validation not be relegated only to the supplementary material but also be described and referenced directly in the main text, since it is central to demonstrating the reliability of the computed perfusion maps. Regarding the external validation issue, I agree with the authors' explanation that suitable datasets are lacking, and I am satisfied with the current approach.

A1: We are happy to move the figure from the supplementary materials into the main manuscript (Fig.1B and 1C), and we have added the following text to the Results section (see page 6):

“High Spatial Correlation Between CPMs and RAPID T-max Maps

We further validated the spatial correlation between CPMs and RAPID T-max in 99 patients. As shown in Fig 1 (B and C), CPMs were highly correlated with RAPID T-max maps (mean Spearman's $\rho = 0.82$, $SD = 0.06$), supporting their validity as an alternative measure of perfusion.”

We have revised the former Figure 1 to include a typical example of CPM and RAPID T-max, and a histogram showing the spatial correlation CPMs and RAPID T-max in 99 patients, creating a multi-panel Figure 1 (A, B and C).

On page 27 in the method section, we have added the following paragraph:

“Validating CPMs against RAPID T-max

To validate our CPMs, we included an additional 91 patients with 4D-CTP acquisitions, in addition to the main cohort. The RAPID-AI software (<https://www.rapidai.com/>, California, USA) was used to derive a time-to-maximum (T-max) map from each 4D-CTP dataset. T-max is a perfusion parameter representing the time for contrast to reach each voxel from the proximal large artery. Furthermore, eight additional patients from the main cohort also had 4D-CTP scans, yielding a total of 99 patients with available T-max maps. Following the same analysis as in our previous study², each exported T-max volume was first converted into a greyscale image and subsequently co-registered to the CPM using SPM12. A Gaussian filter with a kernel size of 10 voxels was then applied to both the exported T-max maps and the CPMs to enhance the visibility of the infarct core⁸⁷. Finally, we calculated Spearman's rank correlation coefficients between the T-max maps and their corresponding CPMs to assess the level of similarity.”

Q2: That said, I would still encourage the authors to perform the exploratory analysis of latent NIHSS dimensions that I originally suggested. While I appreciate their concern about interpretability, even a simple dimensionality reduction (e.g., PCA or factor analysis) would provide valuable complementary insights into the underlying structure of the NIHSS sub-scores. This analysis could remain exploratory and not the main focus of the paper, but including it would enhance the robustness and transparency of the reported findings.

A2: As noted in our previous response, our goal was not to model or predict latent components of the NIHSS, but rather to examine each NIHSS sub-score individually in order to identify neuroanatomically and functionally interpretable substrates, for which NIHSS sub-scores are appropriately structured.

Latent components derived through dimensionality reduction necessarily combine multiple domains, complicating any anatomical interpretation.

We have nonetheless carried out the reviewer's suggestion to perform a principal component analysis (PCA). As shown in Supplementary Figure 15A, the first three principal components explain approximately 70% of the variance (35.74%, 22.05%, and 9.34%, respectively). In Supplementary Figures 15B and 15C, the first principal component (PC01) is shown to capture primarily lateralisation (mainly right motor skills, language, and consciousness); PC02 captures overall stroke severity; and PC03 language and LOC-Question. Note that DLMs of these components would be very hard to interpret: PC01 would elicit neural substrates associated with right motor skills, language, and consciousness; PC02 represents stroke severity; and PC03 reflects language-related areas, but less specifically than the language sub-scores. Note also that a PCA decomposition, as illustrated by the weights in Supplementary Figure 15C, may assign negative values to deficits, which are difficult to interpret biologically with respect to neural function (though not the combination of stroke and functional anatomy, which we are here trying to disentangle).

We can nonetheless illustrate the loading of each component anatomically by applying the weights of each sub-score to its corresponding map and displaying the voxel-wise sum. Since PCA represents a weighted superposition of the original NIHSS sub-scores, we reconstructed the weight superposition mask of PC01 by combining the fifteen inferred neuronal substrates from each NIHSS sub-score, weighted by their corresponding PCA weights. This highlights that each principal component reflects a mixture of different neuroanatomical substrates (Supplementary Figure 15C), making direct interpretation challenging.

Furthermore, we also performed an exploratory study as a secondary analysis of DLM using PC01 as input (see Figure R1 below). The PC01 grey matter (GM) inferred substrate covers the left motor area and the right cerebellum, reflecting the strong correlations of PC01 with the right hand and leg NIHSS sub-scores. The additional regions, such as the thalamus and praecuneus, likely represent secondary but robust correlations with the LOC-command and LOC-question scores. Other regions in the PC01 GM map, including Broca's and Wernicke's areas, reflect correlations with the language NIHSS sub-score (as seen in Supplementary Figure 15B). Because PC01 combines multiple functional domains, the corresponding DLM-inferred substrate represents a mixture of several overlapping networks and therefore cannot be meaningfully interpreted anatomically. Therefore, without prior knowledge of the underlying NIHSS sub-scores, it would be difficult to disentangle these components, rendering the DLM result uninterpretable.

Although we agree with the reviewer about this exploratory analysis, we consider it not central to the main message of this paper. Therefore, we decided to include only the related discussion in the main manuscript (see page 17) and Figure S15 in the supplementary section, while the uninterpretable DLM result (Figure R1, below) is presented only here to avoid potential confusion for readers.

"NIHSS Dimensionality and Spatial-Neuroanatomical Insights

In this study, we analysed NIHSS sub-scores individually rather than modelling latent components, as dimensionality reduction necessarily combines functional domains and can obscure anatomical interpretation. We performed a PCA on the NIHSS sub-scores; the first three principal components (PCs) explained approximately 70% of the variance (Supplementary Fig. 15A). As expected, most PCs represented weighted mixtures of multiple NIHSS sub-scores, with lateralisation, overall severity, and language deficits dominating the first three. We visualised the anatomical loadings of this component by multiplying each of the 15 sub-score maps by their corresponding weights and displaying the resultant voxel-wise sum (Supplementary Fig. 15D), revealing the expected anatomical lateralisation."

Supplementary Figure 15: **(A)** Principal component analysis (PCA) of NIHSS sub-scores. **(B)** Correlation coefficients between each NIHSS sub-score and each principal component. **(C)** PCA weights for each NIHSS sub-score. **(D)** Superposition of all inferred substrate maps weighted by PCA loadings from the first principal component. This PCA-weighted superposition mask was generated as a linear combination of all NIHSS sub-score masks, weighted by their corresponding PCA loadings.

Figure R1: The top panel shows the gray matter (GM) perfusion-deficit map for the first principal component (PC01), which explains 35.74% of the variance. The colour bar indicates the strength of the association between each brain region and the PC01 score. Voxels with weights exceeding 1,273 are considered significant. The bottom panel shows the corresponding white matter (WM) tract disruptions for PC01. The colour bar indicates the normalized magnitude of disruption, scaled from 0 to 1 (MDU: magnitude of disruption unit).

Q3: Overall, I am satisfied with the revisions and responses, but I would like to stress the suggestion about the inclusion of the RAPID validation in the main text and the addition of the exploratory latent analysis before final acceptance.

A3: We thank the reviewer and have added the exploratory latent analysis (see response to Q1).

Reviewer #2 (Remarks to the Author):

The authors have answered my previous question quite well.

Extra questions:

Q1: I am still confused about how the method can be deployed in a clinical setting. The paper could be further strengthened by presenting more quantitative results.

A1: Our CPMs can be generated within ~10 minutes (see page 15) using the current (non-optimized) code, and we expect further reductions in processing time with parallelization and code refactoring. Importantly, CPMs provide mechanistic insight into cerebral blood perfusion directly from routine CT/CTA, without requiring additional scans or radiation exposure. We have now added this validation and merged them into Figure 1, validating CPMs against RAPID T-max maps in 99 patients, which demonstrated a high spatial correlation (mean Spearman's $\rho = 0.82$). We therefore believe that being able to obtain rapid, physiologically interpretable perfusion information within the first few minutes of admission could support clinicians in optimizing treatment decisions in the hyperacute period. Of course, any clinical implementation would require further validation and regulatory approval as a clinical decision-support device, for which this is intended to be a foundational proof-of-concept.

To clarify this, we have added the following paragraph in the discussion section (see page 17).

“Clinical Feasibility and Translation

Our method is designed to be compatible with routine hyperacute workflows. Computed perfusion maps can be generated from standard CT/CTA in approximately 10 minutes using our current (non-optimized) pipeline, with scope for substantially faster processing through parallelization and code refinement. This enables rapid assessment of cerebral perfusion without additional scanning or radiation exposure. While further prospective validation and appropriate regulatory approval are needed before any clinical deployment is considered, the ability to obtain physiologically interpretable perfusion information from initial imaging may support clinical decision-making in the critical hyperacute management period. CPMs may also allow higher fidelity, quantitative characterisation of the vascular state of the brain in advance of treatment that enables greater precision in estimating treatment effects in the context of interventional trials.”

REVIEWERS' COMMENTS:

Reviewer #1 (Remarks to the Author):

Q1: Thank you for addressing my concerns. I am satisfied with the integration of the raised questions and have nothing more to add.

A1: We thank the reviewer for their constructive and insightful feedback and are pleased that our revisions have addressed all concerns.

Reviewer #2 (Remarks to the Author):

Q1: The authors generally address my concerns, but ultimately, we still need stroke clinicians to be involved. How to optimize collaboration between clinicians and this model remains an important issue that should be addressed in future work.

A1: We thank the reviewer for this important point. We fully agree that successful clinical deployment of our approach will require close collaboration with stroke clinicians, both to refine its integration into hyperacute workflows and to appropriately evaluate its utility in real-world settings. Our present study is intended as a methodological and mechanistic foundation. Future work, now being planned with clinical partners across multiple centres, will focus on prospective validation, workflow optimisation, and user-centred integration to ensure clinical applicability. We have expanded the last paragraph of the discussion (before the conclusion) to highlight this point, see page 17-18.

“Establishing the full range of clinical utility of the approach will require careful exploration of the relations between computed perfusion maps and clinical outcomes, and their modulation by the clinical context, facilitated by the large-scale retrospective data the widespread use of CTA in stroke management provides.”